# Population-based heteropolymer design to mimic protein mixtures

Zhiyuan Ruan[1], Shuni Li[2], Alexandra Grigoropoulos[1], Hossein Amiri[3], Shayna L. Hilburg[4], Haotian Chen[1], Ivan Jayapurna[1], Tao Jiang[1,11], Zhaoyi Gu[1,12], Alfredo Alexander-Katz[4], Carlos Bustamante[3,5,6,7,8], Haiyan Huang[2,9] & Ting Xu[1,6,10 ✉]

Biological fluids, the most complex blends, have compositions that constantly vary and cannot be molecularly defined[1]. Despite these uncertainties, proteins fluctuate, fold, function and evolve as programmed[2–4]. We propose that in addition to the known monomeric sequence requirements, protein sequences encode multi-pair interactions at the segmental level to navigate random encounters[5,6]; synthetic heteropolymers capable of emulating such interactions can replicate how proteins behave in biological fluids individually and collectively. Here, we extracted the chemical characteristics and sequential arrangement along a protein chain at the segmental level from natural protein libraries and used the information to design heteropolymer ensembles as mixtures of disordered, partially folded and folded proteins. For each heteropolymer ensemble, the level of segmental similarity to that of natural proteins determines its ability to replicate many functions of biological fluids including assisting protein folding during translation, preserving the viability of fetal bovine serum without refrigeration, enhancing the thermal stability of proteins and behaving like synthetic cytosol under biologically relevant conditions. Molecular studies further translated protein sequence information at the segmental level into intermolecular interactions with a defined range, degree of diversity and temporal and spatial availability. This framework provides valuable guiding principles to synthetically realize protein properties, engineer bio/abiotic hybrid materials and, ultimately, realize matter-to-life transformations.

Whereas molecular precision can be readily achieved inside test tubes, biological fluids are diverse, complex and full of uncertainty[7]. Evolutionarily, the selection of the fittest proteins depends on their surroundings. As a result, natural proteins harmoniously coexist with each other and collectively execute complex tasks with exceptional fidelity amid random fluctuations and external perturbations (Fig. 1a)[8–10]. Information embedded in the sequence space of natural proteins provides the blueprint to design synthetic heteropolymers[11,12] to achieve predictable interplay with biological components as protein substitutes, to holistically recapitulate collective behaviours in protein mixtures and to further gain functions of special proteins while maintaining system compatibility.

We propose that the chemical and sequence characteristics of proteins at the segmental level, as opposed to the monomeric level, is the key factor governing how they transiently interact with neighbouring molecules and the collective behaviours of biological fluids. Experimentally, mimicking the length distribution of blocks containing consecutive residues of the same characteristics in soluble proteins has been proven effective in designing random heteropolymers (RHPs) to stabilize proteins in non-aqueous media as well as to mimic channel proteins to transport protons[13–15]. In both cases, the RHP chains have substantially reduced conformational freedom, being either anchored at polar or non-polar interfaces or spanning a lipid bilayer; and the observed function readouts are from a subpopulation within each designed RHP ensemble. Nevertheless, these results have validated the importance and value of extracting protein sequence information beyond the monomeric level.

In biological fluids, components often interact through random encounters and the whole population needs to be considered. However, when the block-length-based analysis was extended beyond soluble proteins, the block-length distribution broadened (Extended Data Fig. 1 and Supplementary Figs. 1 and 2). Furthermore, the results regarding the block-length distribution contain no information regarding their sequential arrangement within each protein chain. Yet, the effective hydrophobicity of a given block depends on the chemical characteristics of neighbouring ones[14], which affects the probability of it being

[1]Department of Materials Science and Engineering, University of California Berkeley, Berkeley, CA, USA. [2]Department of Statistics, University of California Berkeley, Berkeley, CA, USA. [3]Institute for Quantitative Biosciences-QB3, University of California, Berkeley, CA, USA. [4]Department of Materials Science and Engineering, Massachusetts Institute of Technology, Cambridge, MA, USA. [5]Department of Molecular and Cell Biology, University of California Berkeley, Berkeley, CA, USA. [6]Department of Chemistry, University of California Berkeley, Berkeley, CA, USA. [7]Department of Physics, University of California Berkeley, Berkeley, CA, USA. [8]Howard Hughes Medical Institute, University of California Berkeley, Berkeley, CA, USA. [9]Center for Computational Biology, University of California, Berkeley, CA, USA. [10]Materials Science Division, Lawrence Berkeley National Laboratory, Berkeley, CA, USA. [11]Present address: Department of Chemistry, Xiamen University and The MOE Key Laboratory of Spectrochemical Analysis and Instrumentation, Xiamen, China. [12]Present address: Departments of Chemistry and Biomedical Engineering, Northwestern University, Evanston, IL, USA. ✉e-mail: tingxu@berkeley.edu

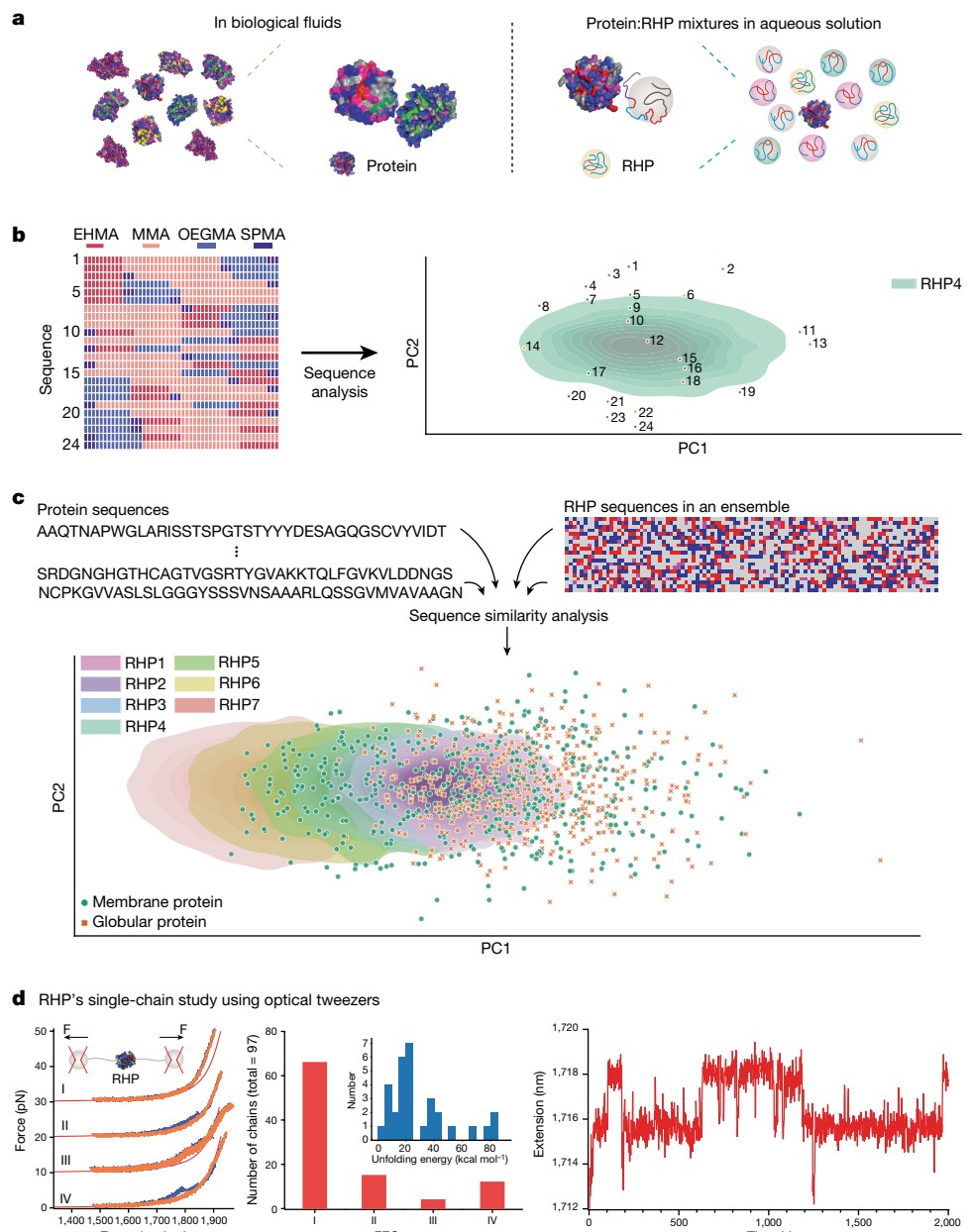

**Fig. 1 | Population-based design to mimic the native environments of proteins. a**, Heteropolymer ensembles are designed to mimic protein mixtures in biological fluids through transient interactions with neighbouring proteins. **b**, The left shows the permutated sequences in descending order of PC2 value from top to bottom. The right shows the projection of 24 permutated sequences with RHP4 composition ratio and RHP4 ensemble onto the PCA space. **c**, 2D sequence analysis of proteins and heteropolymers can be used to compare similarities in their chemical characteristics at the segmental level. 100 membrane protein sequences (green dot) and 100 globular protein sequences (orange cross) (both randomly selected) are shown for clarity. Each shaded area is a univariate distribution of an ensemble of 2,000 RHP chains

using kernel density estimation. **d**, Single-chain study of RHP4 using optical tweezers. The left panel shows four representative types of FEC during pulling (blue) and relaxing (orange): no unfolding signature (type I), discrete rips (type II), smooth shoulders (type III) and a mixture of rips and shoulders (type IV). The red solid curve represents the extensible worm-like chain model for 6 kb DNA. The plots are shifted along the $y$ axis for clarity by a step of +10 pN. The number of RHP4 chains that show each type of FEC and the distribution of unfolding energies for type II–IV chains (inset) are shown in the middle panel. The right panel shows the passive-mode trace reflecting the unfolding and refolding transitions of an RHP4 chain with type IV behaviour.

surface exposed as well as its spatial location within a globular RHP chain. At the whole chain level, the sequential arrangement of blocks will affect the overall chain conformation, intra- and interchain interactions. Here, we developed a two-dimensional (2D) informative sequence analysis to parameterize and visualize both chemical characteristics and sequential arrangement at the segmental level using an autoencoder model (Fig. 1b) and established design rules to modulate RHPs' similarity to proteins as an ensemble. As a test case, we developed a new

library of RHPs as synthetic cytosols capable of modulating RHP–protein and RHP–DNA interactions, and their spatial compartmentalization on microscopic liquid–liquid phase separation (LLPS).

## 2D informative sequence analysis

The model was trained on a library of protein sequences including membrane proteins and globular proteins collected from the

UniprotKB database[16]. Each sequence was first reduced into four types of pseudo-residue: hydrophilic, hydrophobic, very hydrophobic and charged, on the basis of the respective side-chain hydrophobicity of amino acids (Extended Data Table 1)[17]. Initial sequence analysis was also tested using two or eight pseudo-residues (Supplementary Tables 1 and 2). The choice of four pseudo-residues balances the synthetic feasibility of heteropolymers with the accessible diversity and accuracy of segmental interactions (Supplementary Figs. 3–5)[13]. For each protein, the reassigned protein sequence was truncated into a collection of 50-mer segments for analysis (Extended Data Fig. 1). The 50-residue interval was chosen because it captured most short- and long-range residue–residue contacts in native protein conformation. The residue order and local hydrophobicity of each 50-mer were extracted and represented by a low-dimensional vector, which was projected onto a space comprising the first two principal components (PC1 and PC2) from a principal component analysis (PCA)[18,19]. To a first approximation, the PC1 correlates with the apparent hydrophobicity of the 50-mer; the PC2 correlates with the sequential arrangement of different blocks within the 50-mer. A library of hypothetical chains containing the same four blocks arranged in different orders was analysed for illustration. They can be clearly differentiated on the basis of the 2D sequence analysis (Fig. 1b). The PC1's dependence on the PC2 further highlights the importance of considering the block sequence along a chain and its potential to modulate interchain interactions.

The PCA analysis results for known proteins are shown in Fig. 1c and effectively capture key sequence characteristics for both protein families. Membrane proteins have a subpopulation shifted along the PC1 axis compared to that of globular proteins. Thus, the model captures the distinction between the two protein families through their 50-mer segmental hydrophobicity. The large spatial overlap between the two protein families reflects that common protein motifs are present in both protein families. The two protein families were indistinguishable along the PC2 axis and both are more concentrated in the centre region of PC2. It is reasonable to speculate that proteins with alternating short hydrophobic and hydrophilic segments are less prone to aggregation and are more preferable through evolution.

## Design and synthesis of RHP ensembles

Individual heteropolymers cannot fully capture the protein sequence space, and the exact composition of any bio-fluids remains elusive. Thus, a population-based design is required (Supplementary Fig. 6). Individual RHP chains are statistically sequence-defined. They have different monomeric sequences but are statistically similar at the segment level, making them an ideal platform to materialize the segmental information extracted from primary protein sequences[14,20]. A library of RHP ensembles and two diblock heteropolymers (DHP) were designed and synthesized to match the pseudo-residues used in the initial protein sequence analysis (detailed residue assignments are listed in Extended Data Table 1 and Supplementary Tables 1 and 2), using 2–4 out of six selected monomers including methyl methacrylate (MMA), 2-ethylhexyl methacrylate (2-EHMA), 3-sulfopropyl methacrylate potassium salt (3-SPMA), 2-(dimethylamino) ethyl methacrylate (DMAEMA) and oligo (ethylene glycol) methacrylate (OEGMA) with number-average molecular weight ($M_n$) of 300 or 500 Da, respectively. The MMA:EHMA molar ratio was varied in 5 or 10% increments to modulate the distribution of segmental hydrophobicity. Details of all RHPs studied are shown in Table 1.

For each RHP ensemble, 2,000 simulated RHP sequences based on the Mayo–Lewis equation using experimentally determined reactivity ratios and monomer compositions were analysed[20]. This sample size is representative of the ensemble, because the occupied PCA space was observed to converge at roughly 1,500 chains (Supplementary Fig. 6). The segment distribution from proteins and RHP1–7 was projected onto the same PCA space for similarity comparison (Fig. 1c). The occupied

PCA space shifts along the PC1 axis, with some overlapping regions, as the monomer composition varies. RHP ensembles populating the left of the PCA space (lower PC1 value) have more hydrophobic segments than those to the right (Supplementary Fig. 7). The PC2 distribution of RHP ensembles are similar to each other and comparable to those of both protein families. RHP ensembles can be designed to match the segmental diversity of both protein families. There is a sizeable overlap between four RHP ensembles, RHP1–4, with globular proteins; RHP1–7 have some overlap with membrane proteins.

## Probing RHP single-chain conformation

Each RHP ensemble covers a defined range of segmental hydrophobicity and their sequential arrangement. We performed single-chain end-to-end pulling and relaxation studies using optical tweezers[21]. RHP4 shows large overlap with both families of proteins in PCA space and was chosen for single-molecule studies. Results from the first 30 chains are statistically similar to those from a total of 97 chains, indicating that the measured RHP chains should be representative of the ensemble. The RHP ensemble mimics the structural heterogeneity of proteins as commonly seen in biological fluids that contain intrinsically disordered, partially folded and folded states. The force-extension curves (FECs) of 66 out of 97 chains showed no deviation from standard worm-like chain behaviour, typical of non-interacting polymers (Fig. 1d and Supplementary Fig. 8). Four out of 97 chains showed no discrete unfolding 'rip' but a 'shoulder' feature in the force range of roughly 5–10 pN (Supplementary Fig. 9), reflecting a process of rapid, quasi-equilibrium unfolding or refolding of structures that are only marginally stable[22]. Fifteen out of 97 chains showed discrete rips (Supplementary Fig. 10) and 12 out of 97 chains showed a combination of rips and shoulders (Supplementary Fig. 11), indicating cooperative unfolding of mechanically stable structures[22]. When one of the nine RHP chains with a combination of rips and shoulders was subjected to a constant external force (passive mode), a dynamic transition between distinct force-extension states was observed (Fig. 1d). The results indicate that certain RHP subpopulations have reversible folding–unfolding transitions similar to those of proteins[23]. FECs were consistent between several pulling and relaxation cycles of individual RHP chains. Three subpopulations comprising 31 out of 97 chains form structures with stability of 29.0 ± 22.3 kcal mol$^{-1}$ (mean ± s.d.). This value falls well within the energy range associated with reversible conformational changes in proteins (roughly 10–90 kcal mol$^{-1}$)[24]. Together, an RHP ensemble that matches the PCA space of protein mixtures has a defined range of segmental characteristics and can mimic the conformational diversity of proteins.

## PCA overlap governs RHP/protein interplay

The overlapping PCA regions between proteins and each RHP ensemble indicates similarity in their segmental chemical characteristics, which defines the range of interactions during random encounters in biological fluids. We propose that to design RHPs as protein mixture mimics, it is more essential to capture the range of intra- and intermolecular interactions in biological fluids rather than replicating their exact compositions, which remain undefined and fluctuating. Thus, we evaluate the RHPs' similarity to proteins using two tests. First, we probe how each RHP ensemble interacts with membrane proteins during folding post-translation using a cell-free expression platform. Second, we measure how the presence of RHP ensembles in fetal bovine serum (FBS), the most commonly used biological fluid, affects FBS's ability to support cell cultures during storage and thermal denaturation[25].

Three representative membrane proteins, outer membrane protease (OmpT), aquaporin Z (AqpZ) and peptide transporter (PepTso) with a C-terminal fused enhanced green fluorescent protein (eGFP)[26], were selected to cover a broad range in the PCA space and different level

## Table 1 | List of RHPs

| | MMA | OEGMA (500) | EHMA | SPMA | OEGMA (300) | DMAEMA | $M_n$[a] | $M_n$[b] | Đ[c] |
|---|---|---|---|---|---|---|---|---|---|
| RHP1 | 70 | 25 | - | 5 | - | - | 18.4 | 20.7 | 1.4 |
| RHP2 | 65 | 25 | 5 | 5 | - | - | 17.9 | 18.3 | 1.5 |
| RHP3 | 60 | 25 | 10 | 5 | - | - | 18.5 | 18.6 | 1.5 |
| RHP4 | 50 | 25 | 20 | 5 | - | - | 36.3 | 27.3 | 1.3 |
| RHP5 | 40 | 25 | 30 | 5 | - | - | 26.5 | 22 | 1.3 |
| RHP6 | 20 | 25 | 50 | 5 | - | - | 25.6 | 20.8 | 1.3 |
| RHP7 | - | 25 | 70 | 5 | - | - | 31.0 | 24.2 | 1.3 |
| DHP1 | 50 | 25 | 20 | 5 | - | - | 17.3 | 17.9 | 1.1 |
| DHP2 | 50 | 25 | 20 | 5 | - | - | 30.5 | 25.8 | 1.4 |
| RHP8 | 60 | - | 10 | - | 15 | 15 | 15.5 | 10.8 | 1.2 |
| RHP9 | 50 | - | 20 | - | 15 | 15 | 15.5 | 11.8 | 1.2 |
| RHP10 | 20 | - | 50 | - | 15 | 15 | 16.4 | 14.6 | 1.1 |
| RHP11 | - | - | - | - | 10 | 90 | 28.7 | 30.1 | 1.3 |
| RHP12 | 50 | 25 | 20 | - | - | 5 | 22.3 | 27.5 | 1.7 |
| RHP13 | 50 | - | 20 | - | 25 | 5 | 17.3 | 11.0 | 1.2 |
| RHP14 | 70 | - | - | - | 25 | 5 | 16.0 | 11.1 | 1.2 |

[a] Estimated by ¹H-NMR and reported in kDa.
[b] Estimated by gel permeation chromatography using DMF as the mobile phase and reported in kDa.
[c] Dispersity.

of folding success without RHPs during cell-free synthesis (Fig. 2a). Experimentally, the RHP solution concentration was set to 0.2 wt%, well below the RHP's critical overlap concentration (>10 wt% for RHPs studied) to minimize crowding effects. The RHP–protein collision rate is roughly $10^5$ s$^{-1}$ so that the transient RHP–protein interactions are not diffusion-limited on the basis of the translation rate of 10–20 amino acids per second in *Escherichia coli*[27,28] and estimated RHP diffusion rate of roughly 50 μm$^2$ s$^{-1}$. In the presence of RHP1–7, a nearly twofold increase in the AqpZ-eGFP protein yield was observed (Fig. 2b) using $^{35}$S-methionine labelling, and there were improvements in the AqpZ-eGFP's folding status (Fig. 2c,d and Supplementary Fig. 12). Both results are positive indications of RHPs' protein-like behaviour. RHP4 has the highest level of overlapping PCA space with proteins and was the best performer. Similar trends were observed for PepTso with RHP5-6 being the best two performers. OmpT has a fairly good folding status without RHPs (Supplementary Fig. 13); this was attributed to its lower apparent hydrophobicity compared to the other membrane proteins, as seen by its higher PC1 value. RHP1–3 with higher PC1 values were the most effective in mediating OmpT folding, whereas RHP6–7 with lower PC1 values have deleterious effects.

We further probed the interplay between RHP ensembles and biological fluids on thermal denaturation as an accelerated case of how proteins under stress sample different conformations and interact with surrounding molecules. Studies were carried out using FBS, a complex mixture of more than 1,000 proteins and other components[25]. Without RHPs, precipitates formed in the FBS solution within several days when stored at room temperature. When RHPs were added, visual inspection showed a substantial reduction in the formation of precipitates over 1 month without refrigeration. The precipitation was more obvious when the FBS was heated without RHPs for 2.5 h at 52 °C (Fig. 2e), indicative of increased protein denaturation. When different RHP ensembles were compared, RHP4 performed better than RHP7 and RHP2 in stabilizing FBS and was used for subsequent studies (Extended Data Fig. 2). When the thermally treated FBS was used as a growth supplement for in vitro cell culture of NIH3T3 fibroblast cells, there was a 19.1 ± 8.5% (mean ± s.d., *n* = 8) reduction in the cell viability. When

0.5 mg ml$^{-1}$ of RHP4 ensemble was added into FBS before heating, the cell viability was retained at 93.0 ± 12.2% of control FBS without thermal treatment. The cell viability increased to 95.9 ± 10.8 and 104.8 ± 7.8% at RHP4 concentrations of 1 and 2.5 mg ml$^{-1}$, respectively. FBS is a biological fluid commonly used in cell culture, so these studies indicate that RHP ensembles with matching PCA spaces can interact with proteins as if they were proteins themselves. In both studies, we minimized specific factors such as molecular chaperones, osmolytes and crowding effects for protein stabilization[29–31]. These results further confirmed the ability of RHP ensembles to navigate through compositional uncertainties during random encounters.

## RHP segment availability depends on local RHP–protein interactions

Each RHP ensemble includes several segments that span a range of segmental hydrophobicity and sequential arrangements within an RHP chain. Ultimately, the availability of these segments during random encounters governs the apparent protein–RHP interactions. Thus, we probe whether the spatial arrangement of segments can be influenced by their immediate environment and the segmental sequence. Solution small-angle X-ray scattering (SAXS) studies showed that RHP4 in water formed a single-chain nanoparticle, 8.8 nm in average diameter, to bury hydrophobic monomers (Fig. 3a). Proton nuclear magnetic resonance ($^1$H-NMR) studies of RHP4 in D$_2$O showed that the surface-exposed segments are more hydrophilic and mobile; the buried segments are more hydrophobic and have a low tendency to snorkel to the surface of globular RHP chains (Extended Data Fig. 3a), consistent with previous atomistic molecular dynamic simulations[32]. However, from $^1$H-NMR, the full-width at half-maximum (FWHM) accounting for the end-methyl protons of the EHMA side-chain and backbone methyl protons transitioned from being fairly broad to sharp on heating from 25 to 70 °C (Fig. 3b and Extended Data Fig. 3b). In the same temperature range, the FWHM of protons from the end methyl group in the OEGMA side chains remained sharp consistently (Extended Data Fig. 3c). This result indicates buried hydrophobic residues or segments become more flexible and solvated on heating[33]. Adding dimethyl sulfoxide (DMSO) into

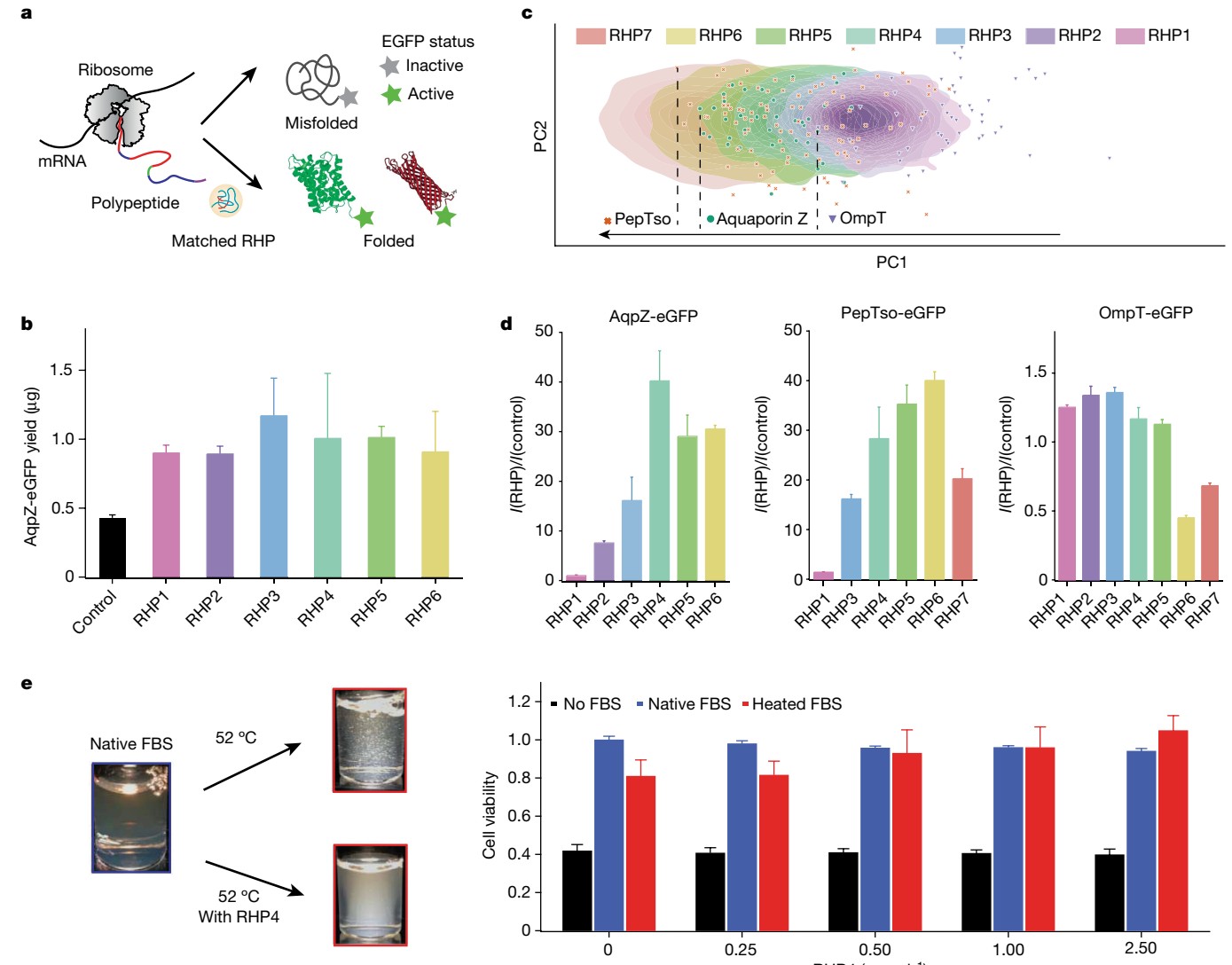

**Fig. 2 | RHP/protein PCA space overlap determines their interplay. a**, Scheme showing RHPs facilitate a top-down conformational sampling process of membrane proteins during protein folding. **b**, Protein yield of AqpZ-eGFP in 8 µl of cell-free reaction in the presence of 0.2 wt% RHPs ($n = 2$). Error bar is 1 s.d. **c**, PCA map showing the segment distributions for both RHP libraries and tested proteins (OmpT (β-barrel), AqpZ (α-helical) and PepTso (α-helical)). **d**, Folding status of AqpZ-eGFP, PepTso-eGFP and OmpT-eGFP in the presence of 0.2 wt% RHPs based on the eGFP fluorescence ($n = 3$). $I$(RHP) and $I$(control) are the fluorescence intensity of tested proteins in the presence and absence of RHP, respectively. Error bar is 1 s.d. **e**, The left shows the presence of RHP4 prevents FBS precipitation on heating. The right shows the cell viability of NIH3T3 as a function of RHP4 concentrations ($n = 8$; the final RHP concentrations in the cell culture media were diluted fivefold from labelled concentrations in the $x$ axis). Error bar is 1 s.d. The cell viability was indirectly measured by metabolic viability-based assays using tetrazolium salts, MTT. At any given concentration of RHP4, RHP4 alone (no FBS), fresh FBS–RHP4 mixture (native FBS) or heated FBS–RHP4 mixture (heated FBS) was fed into the cell culture media.

the $D_2O$ has a similar effect to heating and also causes the proton peak of the EHMA side chains to sharpen (Extended Data Fig. 4).

During random encounters, the molecules in contact with an RHP can locally solvate and lower the energy barrier to surface-expose amphiphilic or hydrophobic RHP segments[34]. No method exists to directly image the RHPs' conformations when they transiently interact with proteins, so an atomistic molecular dynamics simulation was performed. In the simulation, a non-polar hexane nanodroplet was placed near an RHP4 in water. Our simulations showed that an RHP4 globule can be fused and subsequently unravelled at the interface very quickly (roughly 100 ns) (Fig. 3c). This is by stark contrast to frozen RHP4 backbones observed in pure water[32]. There is a substantial conformational rearrangement of RHP at the hexane nanodroplet/water interface, shielding non-polar groups from exposure to water (Supplementary Figs. 14–16). When all studies are considered together, it is reasonable to conclude that an RHP

chain can effectively modulate its side-chain distribution on the basis of the proteins it contacts, and can thus provide matching segments on-demand to assist proteins as they traverse back to their native states.

An RHP chain can be viewed as an equivalent freely jointed chain with segments spanning a range of segmental hydrophobicity (Fig. 3d). The exchange dynamics of each segment depend on its hydrophobicity and length, analogous to the single-chain desorption or exchange kinetics of amphiphilic block copolymers[35]. To understand how the segment length affects RHPs' conformational plasticity, we synthesized two DHPs, $p$(MMA-co-EHMA)-b-$p$(OEGMA-co-SPMA), with the same composition as RHP4, but different molecular weights ($M_n$ (DHP1) of 17.3 kDa and $M_n$ (DHP2) of 30.5 kDa). DHPs have multimodal distributions in the PCA space that account for 50-mers from the $p$(MMA-co-EHMA) block, in-between block or $p$(OEGMA-co-SPMA) block, respectively (Extended Data Fig. 5). The occupied PCA space shifts along both PC1 and PC2

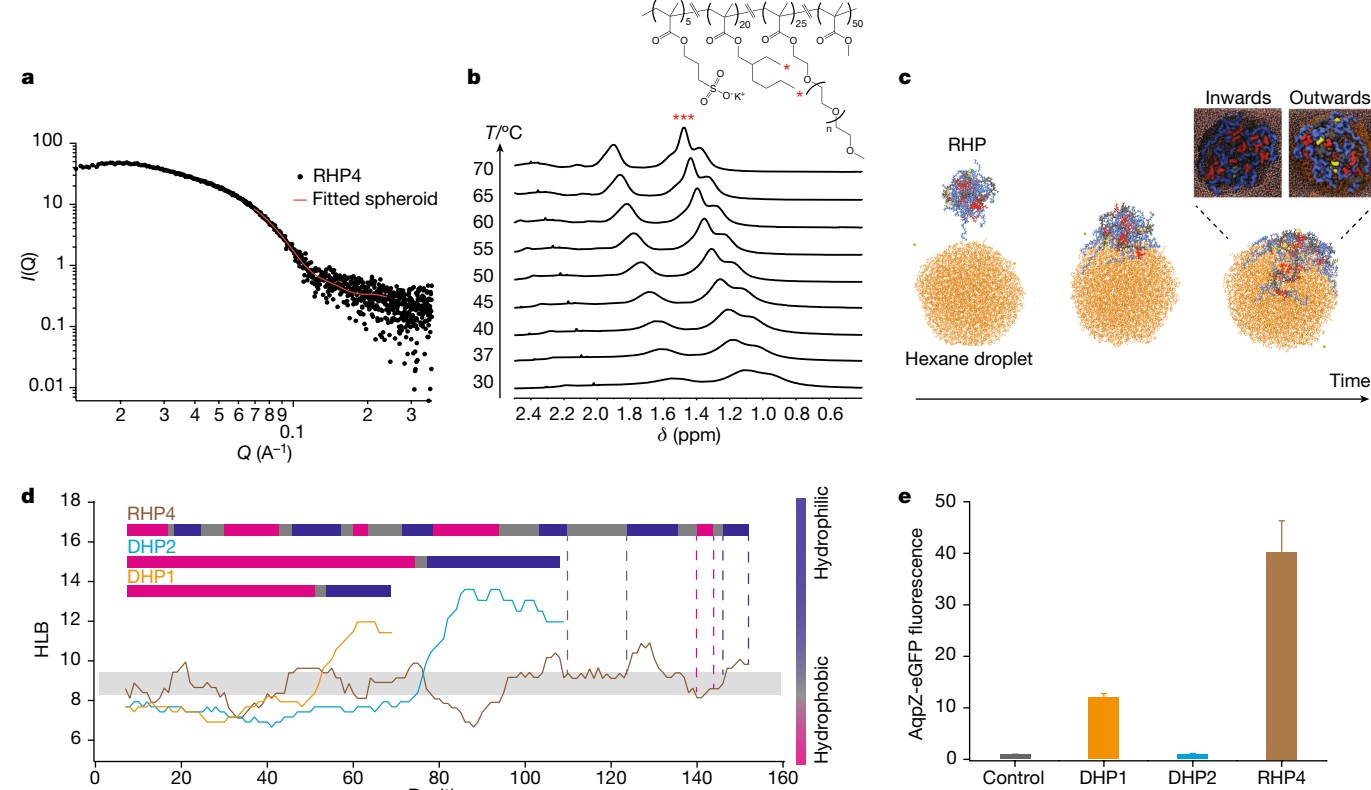

**Fig. 3 | RHPs provide diverse segments with a defined range of segmental hydrophobicity to modulate transient intermolecular interactions. a**, The solution SAXS profile of RHP4 in water at 5 mg ml⁻¹ fitted using a spherical model with a diameter of 8.8 nm. **b**, Part of 1H-NMR spectra of RHP4 as a function of temperature (30–70 °C). **c**, RHP4 adjusts local segmental conformation when exposed to a droplet of hexane, suggesting RHP segments could be available based on the surface of protein in contact. The RHP side-chain configurations at the interface towards the hexane phase ('inwards') and towards the water phase ('outwards') are shown. The system was equilibrated over 100 ns and explicit solvent and counterions are omitted for clarity. Hexane is shown in orange with MMA, OEGMA, EHMA and SPMA in grey, blue, red and yellow, respectively. **d**, Sliding window analysis showing the segmental hydrophobicity along a chain of RHP4, DHP1 and DHP2, respectively. The hydrophobicity of each monomer along a polymer chain was evaluated by the average HLB value of a sliding window. Differences in the diversity of segments are seen between RHPs and diblock copolymers. The block length in copolymers also defines the range of segmental hydrophobicity. **e**, The folding status of eGFP-fused AqpZ in the presence of 0.2 wt% DHP1, DHP2 and RHP4, respectively (n = 3). The corresponding fluorescence intensity is normalized to that measured from the control experiment (no RHP). Error bar is 1 s.d.

axes, leading to a substantial deviation from the protein PCA space. In an assay of AqpZ folding status, DHP1 could increase the eGFP fluorescence intensity during translation, but did not perform as well as its RHP counterparts (Fig. 3e). Negligible enhancement of eGFP fluorescence was detected when DHP2 was used. Dynamic light scattering (DLS) measurements showed that both DHPs have bimodal size distributions with a mean diameter of 56.2 nm for DHP1 (Supplementary Fig. 17) and 145.5 nm for DHP2 (Supplementary Fig. 18), much larger than the 9.1 nm diameter of their RHP counterpart (Supplementary Fig. 19). Thus, the PC2 value is important for predicting the RHPs' tendency to form large assemblies that will compromise the PC1 similarity between RHPs and proteins. The prevalence of short hydrophobic/amphiphilic blocks in RHPs is critical to provide its conformational flexibility, thus ensuring segments' availability.

## Designing new RHP ensembles as cytosol mimics

Biomacromolecules need to function coherently regardless of their own specialties. We propose that the sequence space of proteins encodes how protein mixtures behave beyond individual interactions. Therefore, a population-based design approach for synthetic protein analogues is more meaningful than pursuing molecularly precise formulations. In comparison to the model blends used at present to replicate biological multi-scale phase behaviours, RHP solutions are

more relevant because they access key characteristics such as chemical diversity, compositional complexity and uncertainty[10,36-38]. As a test case, a new RHP library was synthesized to mimic cytosols with common functions including: (1) tunable microscopic LLPS under biologically relevant conditions; (2) the ability to interface and modulate protein folding and stability and (3) compartmentalization of biomolecules such as DNAs to modulate their bioavailability. Designing RHPs to fulfil many functions, each with a high level of biological importance, will test RHPs' similarities to natural proteins and the robustness of population-based design.

RHP8–14 (Table 1) were synthesized using 2–4 of the available monomers: MMA, EHMA, OEGMA ($M_n$ = 300 Da), OEGMA ($M_n$ = 500 Da) and DMAEMA. Two OEGMA monomers with different side-chain lengths and DMAEMA were chosen to implement and modulate temperature-dependent intermolecular interactions[39] to access LLPS under biologically relevant conditions. The RHP monomer ratios were selected to have different levels of overlap in the 2D PCA space with known proteins undergoing LLPS behaviours (from the LLPSDB database[40]). Sequence analysis showed that RHP8–10 overlap with the 2D PCA space of proteins undergoing LLPS, and RHP11 lies outside the protein space (Fig. 4a).

RHP8–10 formed spherical droplets suspended in the buffer at 47 °C for less than 1 min (Fig. 4a,b and Extended Data Fig. 6). The cloud point temperatures of RHP8, 9 and 10 solutions at 1 mg ml⁻¹

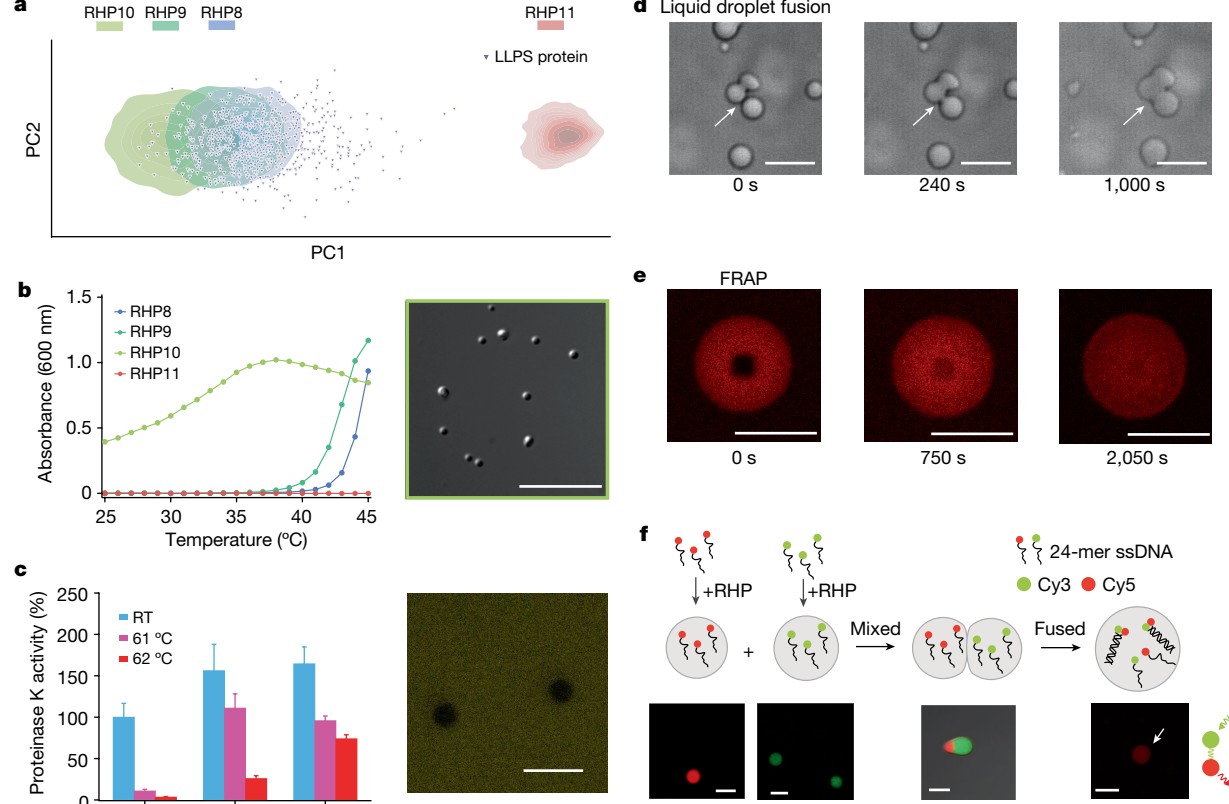

**Fig. 4 | Designing RHP ensembles as synthetic mimics of cytosol. a**, The PCA map of segment distributions for both RHP8–11 ensembles and proteins. **b**, The left shows the temperature-dependent turbidity for RHP8–11 ensembles (1 mg ml⁻¹ in sodium phosphate buffer (50 mM, pH 7.0)). The right shows a differential interference contrast image that RHP10 ensemble forms liquid droplets at 47 °C. **c**, The left shows the normalized hydrolytic activity of ProK before and after heat treatment in the absence or presence of RHPs (*n* = 3).

Error bar is 1 s.d. The original activity of ProK before heat treatment was set to 100%. The right shows the confocal image that the fluorescein-labelled ProK was excluded from RHP14 droplets. RT, room temperature. **d**, Spherical RHP10 droplets fuse into a larger sphere. **e**, FRAP showing the dynamic reorganization of Cy3 dye within an RHP10 droplet. **f**, The confocal images of RHP10 coacervates incubated with 3′-Cy3 ssDNA1 (24 nt; (CAGT)6), 5′-Cy5 ssDNA2 (24 nt; (ACTG)6) or ssDNA1–ssDNA2 duplex. Scale bars, 5 μm.

(sodium phosphate buffer, 50 mM, pH 7.0) were determined to be 43, 40 and <25 °C, respectively. Under the same buffer condition, RHP11 showed no phase separation even up to 70 °C. Thus, matching the PCA space between proteins and RHP ensembles is an effective approach to replicate the collective behaviours of proteins, such as LLPS.

We studied RHP–protein interactions with and without LLPS using RHP12 and RHP13. Both have the same monomer ratio as RHP4, except that SPMA was replaced by DMAEMA and the side-chain length in OEGMA was varied. RHP12 has a cloud point temperature of roughly 67 °C and the cell-free AqpZ-eGFP folding assay at 37 °C confirmed that RHP12 (0.2 wt%) can facilitate AqpZ folding. In comparison, RHP13, which has a cloud point temperature lower than 25 °C, underwent LLPS at 37 °C and showed no effect in the AqpZ-eGFP folding (Extended Data Fig. 7). A control experiment showed that RHP13 did not interfere with eGFP expression or folding post translation. We also assessed how the formation of liquid droplets might affect proteins on thermal denaturation using a common enzyme, proteinase K (ProK). RHP1 has the most PCA space overlap with ProK (Supplementary Fig. 20) and was the best performer as ProK's thermal protectant. RHP1's monomer ratio was adopted to synthesize RHP14 by replacing SPMA with DMAEMA to induce LLPS. In the presence of RHP14, ProK retained 74 ± 5% enzyme activity after being heated at 62 °C for 10 min. In the control experiments without RHPs, ProK lost roughly 97% of its enzymatic activity (Fig. 4c). RHP14 has a cloud point temperature of 33 °C (Extended Data Fig. 8). Confocal studies showed that the fluorescein-labelled ProK

was excluded from RHP14 liquid droplets (Fig. 4c). In comparison, only 25% of ProK activity was retained with RHP1, which showed no LLPS. Together, these results indicated that once the LLPS occurred, the hydrophobic RHP segments and some amphiphilic RHP segments became inaccessible to proteins outside the droplets. These segments are the key to mediating membrane protein–water interactions and assisting membrane protein folding. Their absence reduced the probability of ProK misfolding during thermal denaturation. These results also indicated these hydrophobic or amphiphilic segments provided the driving forces to form the liquid droplets. NMR studies showed no difference in the monomer compositions between the RHPs inside of the liquid droplets and the original RHP ensemble. Thus, it is reasonable to speculate that the interchain interactions within the droplets are dominated by the apparent segmental hydrophobicity within the RHP chains instead of the whole chain.

RHP10 formed many small droplets that ranged from hundreds of nanometres to a few micrometres in size. They diffused, collided with one another and fused together within a few minutes (Fig. 4d). Cyanine 3 amine (Cy3) dye was concentrated in the droplets and used to probe the local viscosity within droplets through fluorescence recovery after photobleaching (FRAP). Nearly complete fluorescence recovery was observed with a characteristic recovery time of 385 ± 15 s (Fig. 4e and Extended Data Fig. 9). This time scale is comparable to that of MEG-3 proteins (128–384 s) in P granules[41,42]. The RHP droplets have a fluidity similar to membraneless organelles, and offer a promising path towards their synthetic mimics.

We propose that the design rules used here to modulate RHP–protein interactions can also be applied to tailor how RHPs interact with DNA. Fundamentally, RHPs enable us to evaluate energetic competition between a wide range of interactions, going beyond the solely electrostatic interactions commonly studied in coacervation. When 1 μM of 24-mer single-stranded DNA (ssDNA, 24 nt) was added to the RHP10 solution (1 mg ml$^{-1}$, sodium phosphate buffer, 50 mM, pH 7.0), ssDNA was selectively encapsulated inside the liquid droplets at 37 °C (Fig. 4f). Similar results were observed for a 24-base pair fluorescence resonance energy transfer (FRET)-pair labelled double-stranded DNA (dsDNA). FRET was observed inside the liquid droplets, indicating the absence of dsDNA dissociation. When liquid droplets carrying complementary FRET-pair labelled ssDNA were mixed, these droplets fused. Subsequently, complementary ssDNA strands formed duplexes and emitted FRET over the course of 25 min. These results confirmed that the RHP–DNA interactions are strong enough for selective partition but do not interfere with specific DNA pairing interactions. Thus, designed RHPs can capture the range of intra- and intermolecular interactions required to be compatible with many biological processes occurring inside of cytosol, making them viable building blocks towards bio- or abiotic materials.

The present studies clearly demonstrate the feasibility of designing heteropolymers as an ensemble to mirror protein mixtures in biological fluids despite the inexact formulations of these complex blends. The segmental sequence information of proteins can guide statistic sequence control in RHPs to holistically replicate protein functions. Being orthogonal to molecular-precision-driven design, the population-based approach is actually advantageous to navigate through chemical diversity and unpredictable fluctuations abundant in protein's native environments and, ultimately, to interface synthetic materials and biological systems seamlessly.

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

# Methods

## Materials

Chemicals were purchased from Sigma Aldrich Chemical Co. or Fisher Scientific International, Inc. unless otherwise noted. Water was purified by a Milli-Q water filtration station (18.2 $\Omega M \, cm^{-1}$) before use. Azobisisobutyronitrile (AIBN) was recrystallized from ethanol before use. Inhibitors were removed using cryodistillation (MMA, EHMA) or by passing through a short column of neutral alumina (ethylene glycol methyl ether methacrylate) ($M_n$ roughly 500 g mol$^{-1}$). 3-SPMA (98%) before polymerization. Ethyl-2(phenylcarbanothioylthio)-2-phenylacetate, trioxane (internal standard for 1H-NMR analysis) and dimethylformamide (DMF, solvent) were used without further purification. Cell-free transcription/translation system (PURExpress) was purchased from New England BioLabs Inc. Plasmid Aqpz-GFP was a gift from D.L. Minor, Jr. (University of California, San Francisco). Plasmid pWaldoGFPe_PepTSo was a gift from S. Iwata and S. Newstead (Addgene plasmid no. 58334). pET28-OmpT was a gift from N. Kelleher (Addgene plasmid no. 68862). Anti-GFP antibody (B2) was purchased from Santa Cruz Biotechnology. Goat antimouse alkaline phosphatase secondary antibody conjugate was purchased from Bio-Rad Laboratories. End-modified oligonucleotides were purchased from IDT Co. ProK was purified by a desalting column before use. Dulbecco's Modified Eagle Medium (DMEM) and cell culture plates were purchased from Corning Inc. The FBS and phosphate buffered saline were purchased from Gibco Inc. The 3-(4,5-dimethylthiazol-2-yl)-2,5-diphenyltetrazolium bromide (MTT) and pancreatin were purchased from Invitrogen Co. The NIH3T3 cell line was provided by the UCB Cell Culture Facility.

## RHP synthesis

The synthesis of RHPs was carried out using reversible addition-fragmentation chain-transfer polymerization as in our previous work. Polymerization solutions were prepared by mixing the requisite amounts of reagents and added into 25 ml glass Schlenk. The solutions were subject to three freeze–pump–thaw cycles before the Schlenk was placed into the oven. The polymerization solutions were held at 70 °C, typically for 4–8 h. Global monomer conversion was determined by $^1$H-NMR on crude reaction mixtures in DMSO-d$_6$ with trioxane as an internal standard. The polymerization solution was then precipitated by dropwise addition to rapidly stirred pentane. The resultant polymer was then dissolved in water and transferred to a 2,000 molecular weight cut-off (MWCO) dialysis bag and dialysed against distilled water for 3 days. The purified polymer was then subject to lyophilization.

## Diblock copolymer p(MMA-co-EHMA)-b-p(OEGMA-co-SPMA) synthesis

An example of the preparation of $p$(MMA-r-EHMA)$_{53}$-b-$p$(OEGMA-r-SPMA)$_{22}$ ($M_n$ (NMR) = 17,327 g mol$^{-1}$) is given.

**p(MMA-r-EHMA) macro-CTA synthesis.** MMA (1,072.71 mg, 10.71 mmol), 2-EHMA (849.86 mg, 4.29 mmol), CTA (2-cyano-2-propyl-4-cyanobenzodithioate, 32.13 mg, 0.13 mmol), AIBN (2.14 mg, 0.013 mmol) and DMF (1 ml) were mixed in a 20 ml glass vial. The reaction mixture was deoxygenated by N$_2$ flow for 10 min, placed in an oven and allowed to react at 70 °C, typically for 3 h. After polymerization, the crude product was reprecipitated in pentane three times, redissolved in tetrahydrofuran (THF) and transferred to a clean vial. The solvent was removed by evaporation under reduced pressure. The purified polymer was then dried in vacuo overnight.

**p(MMA-co-EHMA)-b-p(PEGMA-co-SPMA) synthesis.** OEGMA (1,027.41 mg, 2.05 mmol), 3-SPMA (101.29 mg, 0.41 mmol), $p$(MMA-r-EHMA)$_{53}$ macroRAFT agent (335.743 mg, 0.0476 mmol,

$M_n$ = 7,056 g mol$^{-1}$), AIBN (0.781 mg, 0.00476 mmol) and DMF (1 ml) were added and mixed in a 20 ml glass vial. The reaction mixture was deoxygenated by N$_2$ flow for 10 min, placed in an oven and allowed to react at 70 °C for roughly 3 h. After polymerization, the crude product was reprecipitated in pentane three times, redissolved in THF and dialysed against a 12:1 mixture of THF and water for 3 days. The solvent was removed by evaporation under reduced pressure. The purified polymer was then subject to lyophilization.

## Polymer characterization

Molecular weight distribution curves, number-average ($M_n$) and weight-average ($M_w$) molar mass and dispersity ($Đ = M_w/M_n$) of copolymers were measured by gel permeation chromatography using an Agilent 1260 Infinity series instrument outfitted with two Agilent Poly-Pore columns (300 × 7.5 mm). DMF with 0.05 M LiBr was used as the eluent at 0.7 ml min$^{-1}$ at 50 °C. The columns were calibrated against Poly (ethylene glycol) standards or by light scattering. Analyte samples at 2 mg ml$^{-1}$ were filtered through 0.2 µm polytetrafluoroethylene membranes (VWR) before injection (20 µl). $^1$H-NMR spectra were recorded in DMSO-d$_6$ and D$_2$O on a Bruker Avance 400 spectrometer (400 MHz) using a 5 mm $Z$ gradient BBO probe or on a Bruker Avance AV 600 spectrometer (600 MHz) using a $Z$ gradient Triple Broad Band Inverse detection probe.

## Cell-free protein synthesis

Cell-free protein synthesis was carried out according to PURExpress manual with modifications. In 25 µl of reaction, RHP of the desired amount was added to the ribosome solution and incubated on ice for 30 min. The polymer/ribosome samples were mixed with solution A and B containing all other required components including enzymes, RNAs, energy and nutrient molecules. Next, 20 units of RNase inhibitor and 200 ng plasmids for AqpZ-eGFP, PepTso-eGFP and OmpT-eGFP were added, and the mixtures were incubated at 37 °C for 4 h to complete membrane protein synthesis. The resulting products were stored at −20 °C.

## Kinetics of protein synthesis and western blot analysis

Kinetics of cell-free synthesis were monitored by measuring the fluorescence intensity of the C-terminal GFP tag on a Tecan I-control infinite 200 plate reader. Cell-free protein synthesis mixtures were incubated in a 384-well plate at 37 °C, and the GFP fluorescence (excitation/emission 488 nm/520 nm) was recorded over 4 h. For the western blot, proteins in the cell-free protein synthesis samples were first separated by SDS–PAGE. Protein mass standards were used (Spectra multicolour broad range protein ladder, Thermo Fisher). The proteins were then transferred to a polyvinylidene difluoride membrane. Multicolour protein mass standards were visible on the membrane after successful transfer. The membrane was then blocked with 3% BSA, incubated with 1:2,000 mouse anti-GFP antibody and washed. Following incubation with 1:4,000 alkaline phosphatase conjugated goat antimouse secondary antibody, band colours were developed by using 5-bromo-4-chloro-3-indolyl phosphate and nitroblue tetrazolium.

## Protein yield determination

The protein yield of cell-free synthesis was determined by measuring the $^{35}$S-Methionine incorporated samples in a scintillation counter. EasyTag$^{35}$S-Methionine (7.2 pmol) was added in the 24 µl of cell-free synthesis solution, and the mixtures were incubated at 37 °C for 4 h to incorporate $^{35}$S-Methionine into the proteins. Then 8 µl of the labelled cell-free reaction was mixed with 100 µl of 1 M NaOH and incubated at room temperature for 10 min. Next, 800 µl cold TCA/CAA mix (25% trichloroacetic acid/2% casamino acids) was added to the sample. The sample was briefly vortexed, then incubated on ice for 5 min. The precipitated proteins were collected on

a paper filter by vacuum filtration and measured in a scintillation counter.

## Hydrolytic activity assay for ProK

The hydrolytic activity of ProK with different RHPs was measured by monitoring the absorbance at 410 nm to detect $p$-nitroaniline, which is released after the hydrolysis of 4-nitrophenyl butyrate. For the thermal denaturation assay, a mixture solution containing 0.07 mg ml$^{-1}$ ProK and as required, 2 mg ml$^{-1}$ RHP in sodium phosphate buffer (50 mM, pH 7.0) was incubated at elevated temperature for 30 min. The sample containing 10 μl of the aforementioned solution, 1 μl of 4-nitrophenyl butyrate (50 mM in methanol) and 239 μl of Tris buffer (50 mM, pH 8.0) was analysed in absorption at 410 nm.

## Cell culture

The NIH3T3 cell line was received frozen. The cells were thawed, diluted in DMEM (Gibco, 4.5 g l$^{-1}$ glucose, L-glutamine, sodium pyruvate, 10% FBS) and incubated at 37 °C and 5% CO$_2$. NIH3T3 cell line was divided every 3 days.

## MTT assay

RHP was added to FBS at concentrations of 0.25, 0.5, 1 and 2.5 mg ml$^{-1}$. The mixed solution was incubated at 52 °C for 2.5 h. The supernatant was collected after centrifugation (12,000$g$, 10 min) and diluted by a factor of 5 in DMEM (Gibco, 4.5 g l$^{-1}$ glucose, L-glutamine, sodium pyruvate, denoted SolA). NIH3T3 cells were seeded in 96-well plates at 10,000 cells per well (culture volume 100 μl per well) and incubated at 37 °C and 5% CO$_2$ for 16 h. The cells were then fed with SolA and incubated at 37 °C and 5% CO$_2$ for 24 h. The cell culture media was removed and 50 μl of MTT reagent solution (0.5 mM in DMEM, Gibco, phenol red free, 4.5 g l$^{-1}$ glucose, L-glutamine, sodium pyruvate) was added per well. The plate was incubated at 37 °C and 5% CO$_2$ for 30 min. Next, 150 μl of DMSO was added per well and the plate was shaken to dissolve formazan completely. The absorbance at 570 nm was recorded using Infinite M200 microplate reader (Tecan).

## Differential scanning calorimetry (DSC)

DSC measurements were implemented using the MicroCal VP-DSC system (Malvern Panalytical). All samples were prepared in 50 mM Sodium Phosphate buffer (pH 7.0) with 0.28 mg ml$^{-1}$ ProK and as required, 1.5 mg ml$^{-1}$ RHP. The samples were degassed for 8 min using the MicroCal ThermoVac (Malvern Panalytical) before insertion into the sample cell. When the pressure in the sample holder had stabilized at roughly 28 psi, the cells were cooled to 10 °C, followed by a 15 min wait time. The thermograms were produced by measuring the difference in heat capacity between the sample solution and the reference buffer solution as the temperature was increased from 10 to 90 °C at a rate of 1 °C min$^{-1}$. Following the retrieval of the thermograms, the buffer–buffer reference thermograms were subtracted using OriginLab software, as was any linear baseline observed.

## DLS

DLS measurements were conducted on a Brookhaven BI-200SM Light Scattering System at a 90° scattering angle. The concentration for each measurement was 5 mg ml$^{-1}$ of RHP or 0.5 mg ml$^{-1}$ of DHP.

## Diffusion-limited collision rate estimation (d$n$/d$t$)

$$\mathrm{d}n/\mathrm{d}t = j(r)4\pi r^2 = 4\pi \, \mathrm{Dr}c_\infty = k_{\mathrm{on}}c_\infty$$

where $n$ is the number of molecules, $t$ is time, $j$ is the flux, $r$ is the radius, D is the diffusion coefficient, r is the radius of the protein of interest, $K_{\mathrm{on}}$ is the polymer-protein association rate, and $c_\infty$ is the bulk concentration of polymers.

As proteins are around 4 nm in diameter,

$$k_{\mathrm{on}} = 4\pi \mathrm{Dr} = 4\pi \times 50 \, \mu\mathrm{m}^2/\mathrm{s} \times 2 \times 10^{-3} \, \mu\mathrm{m} \times 6.02$$
$$\times 10^{23} \, \text{molecules per mol} \times 10^{-15} \, \mathrm{l} \, \mu\mathrm{m}^{-3} \approx 10^9 \, \mathrm{s}^{-1} \mathrm{M}^{-1}.$$

At 2 mg ml$^{-1}$ of RHP4 ($M_\mathrm{n} = 36$ kDa),

$$\mathrm{d}n/\mathrm{d}t = k_{\mathrm{on}}c_\infty = 10^9 \, \mathrm{s}^{-1} \mathrm{M}^{-1} \times 2 \, \mathrm{g} \, \mathrm{l}^{-1}/(3.6 \times 10^4 \mathrm{g} \, \mathrm{mol}^{-1}) \approx 10^5 \, \mathrm{s}^{-1}.$$

## Critical overlap concentration of RHP4

$$c^* \cong 3M/4\pi N_\mathrm{A} R_\mathrm{g}^3 \cong 17 \, \mathrm{wt\%}.$$

where $c^*$ is the critical overlap concentration, M is the average molecular weight of the polymer, $R_\mathrm{g}$ is the radius of gyration of the polymer, and $N_\mathrm{A}$ is Avogadro's number.

## SAXS

SAXS was carried out at beamline 8-ID-E at the Advanced Photon Source, Argonne National Laboratory. Samples were dissolved in water at a range of concentrations from 0.2 to 2 wt%. Samples were measured in 2 mm boron-rich thin-walled capillary tubes and subject to several short exposures (5 s for each time). 2D scattering results were azimuthally averaged to produce one-dimensional SAXS profiles. Superimposable profiles were averaged and then subtracted from the background data. The radius of gyration ($R_\mathrm{g}$) of an RHP was obtained from the Guinier plot by fitting the scattering to the following equation, where q is the scattering vector, $I$(q) is the scattering intensity at q, and $I$(0) is the intensity of incident beam:

$$\ln(I(\mathrm{q})) = \ln(I_0) - (R_\mathrm{g}^2/3)q^2$$

## Molecular dynamics simulation

The final frame from an equilibrated sequence (sequence 6) used in previous work[32] was extracted and solvated in a system containing a pre-equilibrated hexane droplet, four potassium counterions and 44,996 water molecules. The droplet is composed of 1,503 hexane molecules and was formed with 63,898 water molecules through 2 ns of equilibration and 40 ns of production simulation. The combined system was equilibrated for 2 ns and then run at production setpoints for 100 ns. All simulations followed the methods and parameters used in previous work, adopting the monomer parameterization methods for hexane molecules. VMD was used for visualization of the resulting trajectories[43]. Solvent accessible surface area (SASA) calculations were performed with Amber19's LCPO default parameters[44]. SASA for the hexane phase is taken as the SASA for the trajectory stripped of all non-polymer atoms (total RHP SASA) minus the SASA for the as-is trajectory (water accessible RHP SASA).

## Synthesis of oligonucleotide-RHP conjugate

Buffer A (storage buffer): 20 mM sodium phosphate, pH 7.83
Buffer B: 200 mM sodium phosphate, 300 mM NaCl, pH 7.24
Buffer C: 100 mM sodium phosphate, 1,500 mM NaCl, pH 7.24
DNA1: GTCGCTCTCTCATGCAGAATCCCA, 1 mM in buffer A
DNA2: CTGCTGGGGCAAACCAGCGTGGAC, 1 mM in buffer A

**Synthesis of RHP with dual end-functional groups of –SH and –N$_3$.** An azido-modified chain-transfer agent (2-(dodecylthiocarbonothioylthio)-2-methylpropionic acid 3-azido-1-propanol ester) was used to synthesize RHP (RHP-N$_3$). Before conjugation, 1 μl of hydrazine was added into 200 μl of RHP solution (20.30 mg, in THF). The mixture was placed on a shaker for 0.5 h. RHP was purified by an Amicon-3K ultrafilter (3,000 MWCO) using de-ionized water six times.

**Synthesis of DNA1-RHP conjugate through thiol-maleimide addition.** Here, 1.5 mg of sulfo-SMCC[45] was dissolved in 100 μl of $H_2O$ (heated up to 50 °C for clear solution), followed by the addition of 100 μl of buffer B. Then 10 μl of 3′-amine-modified DNA1 was added. The solution was placed on a shaker for 2 h at room temperature. The excess sulfo-SMCC was removed by an Amicon-3K ultrafilter (3,000 MWCO) using buffer B six times. Then, 5 nmol of purified sulfo-SMCC-DNA1 (90 μl in buffer C) was mixed with roughly 2 mg of RHP (10 μl in DMF). The reaction was placed on a shaker overnight at room temperature. After the reaction, the excess oligonucleotide was removed by Amicon-3K by washing with de-ionized water six times (Supplementary Fig. 21).

**Synthesis of DNA1-RHP-DNA2 conjugate through azide-alkyne cycloaddition.** Here, 10 μl of 5′-hexynyl-modified DNA2 (1 mM in buffer A) was mixed with 4 μl of DNA1-RHP conjugate (roughly 1 nmol), followed by the addition of 17 μl of DMF. Then 10 μl of fresh click-reaction solution (10 mM $CuSO_4$, 50 mM TBTA, DMF/$H_2O$ = 1:1) and 3 μl of sodium ascorbate (100 mM in water) were added. The mixture was placed on a shaker overnight at room temperature. The excess oligonucleotide was removed by an Amicon-3K ultrafilter (3,000 MWCO) using water six times.

### Single-molecular force spectroscopy

Single-molecule force-extension measurements were carried out in a dual-trap optical tweezers instrument equipped with a 1,064 nm trapping laser[46] at a trap stiffness of 0.1 to 0.2 pN nm$^{-1}$. First, biotinylated double-stranded (3 kilobases (kb)) DNA handles with 24 nucleotide 5′ or 3′ terminal single-stranded overhangs complementary to DNA1 or DNA2, respectively, were deposited on streptavidin-coated polystyrene beads (1 μm). Individual RHP chains were captured between trapped beads by hybridization of their conjugated DNA to the overhangs in buffer containing 20 mM Tris·HCl (pH 7.2), 100 mM KCl, 10 mM $MgCl_2$ and 5 mM beta-mercaptoethanol. Pulling and relaxing force ramps were performed at a rate of 100 nm s$^{-1}$ from less than 1 pN to up to 25 pN and repeated several times for each molecule. Force-extension data were collected at 667 Hz. For molecules that showed a rip in their FEC, passive-mode (constant trap position) measurement was also performed. Unfolding work was calculated by integrating each FEC and subtracting the contribution of the worm-like-chain behaviour of bare 6 kb handle DNA and unfolded RHP.

### RHP sequence generation

An in-house sequence simulator called Compositional Drift[20] was used to generate 15,000 chains for each RHP ensemble (DP = 50).

### Training dataset

To train the sequence autoencoder, 30,000 membrane protein sequences and 30,000 globular protein sequences with 50% identity threshold were collected from the UniProt database[16]. Sequences with uncommon amino acids were discarded. Each protein was reduced into four monomer codes. The assignment of each residue to its monomer equivalent is listed in Extended Data Table 1 and Supplementary Tables 1 and 2. For each protein sequence, a set of consecutive protein motifs were collected by moving a 50-residue long window over each sequence with 15-residue long step size, resulting in a total of 1,046,845 training sequence motifs.

### Latent variable model

An in-house latent variable model was developed to perform sequence dimensionality reduction and to learn protein/RHP sequence representations. The model was implemented on the basis of a typical autoencoder architecture with an extra regression module to force the latent space to learn sequence hydrophilic-lipophilic balance (HLB) distributions. Each protein sequence motif (in its RHP equivalent form) was first one-hot encoded and then passed through the encoder. The encoder embedded each sequence into a 16-dimensional latent vector, $z$, which was then fed into two parallel branches: the decoder and the regressor. The decoder intended to reconstruct the input sequence from $z$, whereas the regressor predicted the HLB values of an input sequence from $z$. The loss function was designed to be a weighted sum of the departure (for example, cross entropy) of the original and reconstructed sequence and the mean squared error of predicted and true HLB values. A meaningful low-dimensional latent space that captures both sequential features and HLB distributions was obtained by optimizing the reconstruction and regression loss together.

Both the encoder and the decoder were implemented with simple multilayer perceptrons. Each had three fully connected layers with 256, 128 and 64 hidden units. The regression module had two fully connected layers with 16 hidden units. Rectified linear unit non-linear functions were used throughout the network, except that Sigmoid activation was used in the output layer of the decoder. The model was trained using the ADAM optimizer with a learning rate of 0.001. A learning rate scheduler was used to reduce the learning rate when the validation loss stopped improving. All model hyperparameters were optimized with Weights and Biases[47]. We also implemented a LSTM-based autoencoder, which demonstrated similar performance as the simpler autoencoder variant model.

### Coacervation

The lyophilized RHP was dissolved in Mill-Q water (30 mg ml$^{-1}$). Then 1 μl of RHP solution was mixed with 29 μl of sodium phosphate buffer (50 mM, pH 7.0). The coacervation was triggered by incubating the solution at 47 °C. The milky colour appeared in less than 1 min and coacervation was confirmed by bright-field microscopy. For ssDNA partitioning, 1 mg ml$^{-1}$ RHP and 1 μM ssDNA (50 mM sodium phosphate buffer, pH 7.0) were incubated at 37 °C for 10 min before imaging. Before the dsDNA partitioning, two complementary ssDNA (5 μM for each) were mixed and heated at 95 °C for 2 min, then immediately cooled at 4 °C for 5 min. The resulting dsDNA solution were diluted to 1 μM and mixed with 1 mg ml$^{-1}$ of RHP. The solution was incubated at 37 °C for 10 min before imaging.

The sequence of ssDNA is shown below:
/5Cy5/ACTGACTGACTGACTGACTGACTG;
CAGTCAGTCAGTCAGTCAGTCAGT/3Cy3Sp/.

### Microscopy

The differential interference contrast microscopy was performed using a Zeiss AxioImager M2 microscope equipped with a Zeiss Plan-Neofluar Ph3 ×100/1.30 oil-immersion objective and QImaging Retiga 1350EX 1.4MP Monochrome CCD Microscope Camera. The DNA partitioning, RHP droplet fusion and FRAP was imaged using a Zeiss LSM 880 FCS confocal microscope with X-Cite 120LED illumination system and GaAsP photon counting detector. For ssDNA partitioning, illumination was provided by a laser with the wavelength of 514 nm (for Cy3 or Cy3-Cy5 FRET pair) or 633 nm (for Cy5). Images were averaged from eight consecutive images and were of the format 1,024 × 1,024 pixels at an 8-bit depth. RHP droplet fusion was imaged every 2 s under transmitted light in a format of 512 × 512 pixels at an 8-bit depth.

### FRAP

Here, 50 μM of Cy3 amine (Lumiprobe, 410C0) was used to probe FRAP within the RHP droplets. The pinhole size was set to 1.5 μm section. FRAP measurement was performed by selecting a thin square region of interest in the centre of a droplet, bleaching with the 488 nm laser line at 4% for 20 s. The photobleached droplet was imaged every 5 s in a format of 512 × 512 pixels at a 12-bit depth (four repetitions). The FRAP analysis was carried out using Fiji software. Before photobleaching, the fluorescence intensity from assigned bleached area and unbleached area was denoted as $I_b^{before}(t)$ and $I_u^{before}(t)$, respectively. $I_b^{after}(t)$ and $I_u^{after}(t)$ represented

the fluorescence intensity from bleach and unbleached area at time $t$ after photobleaching. All fluorescence intensities were normalized by the area. The centre-to-ratio was denoted as $r^{before} = I_b^{before}(t)/I_u^{before}(t)$, $r^{after} = I_b^{after}(t)/I_u^{after}(t)$. Thus, the recovery rate ($R$) was calculated by

$$R = (r^{after}(t) - r^{after}(0))/(r^{before} - r^{after}(0)).$$

The recovery rate curve was fitted by an exponential function:

$$R = A(1 - e^{t/\tau})$$

where $A$ and $\tau$ represent the maximal recovery rate and recovery half time, respectively.

## Data availability

All data needed to evaluate the conclusions in the paper are present in the paper and/or the supplementary materials. For reproduction purposes, the raw data used to generate the figures are available from the Dryad Digital Repository (DOI:10.6078/D1KH8R ).

## Code availability

Source code and input scripts supporting this work are available at https://github.com/Shunili/AE-RHP.

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

**Acknowledgements** The work was supported by the US Department of Defense (DOD), Army Research Office, under contract no. W911NF-13-1-0232, Defense Threat Reduction Agency (DTRA) under contract no. HDTRA1-19-1-0011, the National Science Foundation under contract no. DMR-2104443, the US Department of Energy, Office of Science, Office of Basic Energy Sciences, Materials Sciences and Engineering Division, under contract no. DE-AC02-05-CH11231 (KC3104) and the Alfred P. Sloan Foundation (grant no. G-2021-16757). Z.R. is supported by the Kavli Energy NanoScience Institute through the Kavli ENSI Philomathia Graduate Student Fellowship Program. Scattering studies were done at Advanced Photon Source and use of the Advanced Photon Source was supported by the US Department of Energy, Office of Science, Office of Basic Energy Sciences, under contract no. DE-AC02-06CH1135.

**Author contributions** T.X. conceived the idea and guided the project. Z.R. and T.J. performed cell-free synthesis of membrane proteins. Z.R. and A.G. performed thermal denaturation of globular enzymes. S.L., I.J., Z.R. and H.H. performed sequence analysis. H.A. and C.B. performed optical tweezers analysis. S.H. and A.A.-K. performed all-atom simulation studies. Z.R. and Z.G. synthesized and characterized the RHPs and DHPs. H.C. performed the cell study. A.G. and H.C. performed the confocal study. All authors participated in writing the manuscript.

**Competing interests** T.X., H.H., Z.R. and S.L. have a pending PCT patent application. The rest of the authors declare no competing interests.

**Additional information**
**Correspondence and requests for materials** should be addressed to Ting Xu.

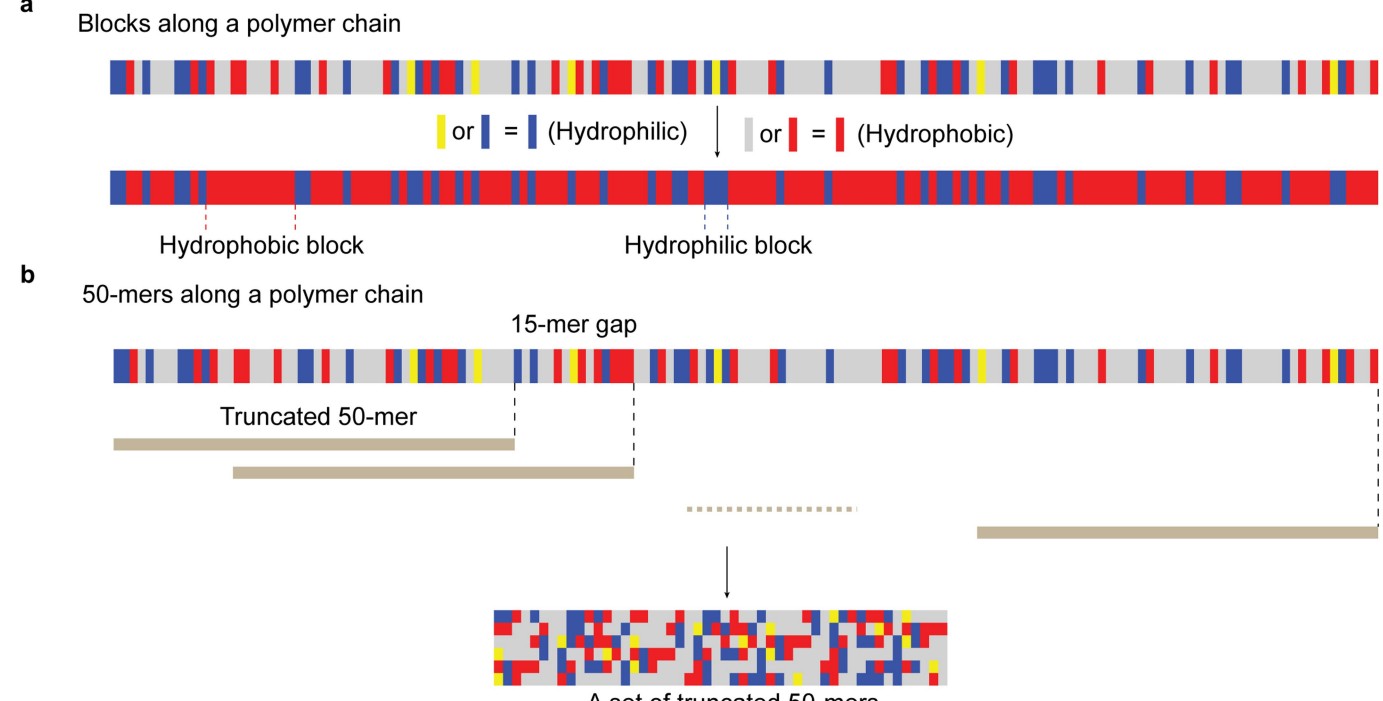

**a**

Blocks along a polymer chain

| or | = | (Hydrophilic)    | or | = | (Hydrophobic)

Hydrophobic block          Hydrophilic block

**b**

50-mers along a polymer chain

15-mer gap

Truncated 50-mer

A set of truncated 50-mers

**Extended Data Fig. 1 | Blocks and 50-mers along a polymer chain.** (a) Each monomer was reassigned to one of two pseudo-monomers (hydrophobic vs. hydrophilic) based on monomer's hydrophobicity. A block comprises consecutive monomers of a single type. (b) A polymer chain was truncated into a set of 50-mers.

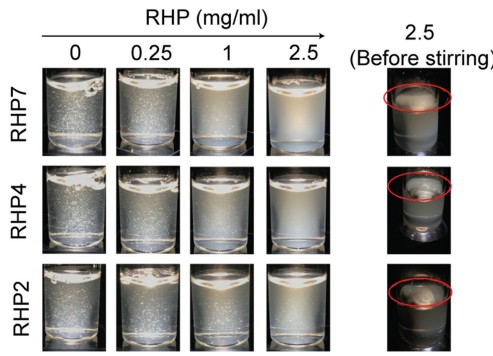

**Extended Data Fig. 2 | The FBS solution with different RHP ensembles after incubating at 52 °C for 2.5 h.** The FBS solution forms a thin film (like "milk skin") at the air-water interface after thermal treatment (red circle). RHP4 exhibits the weakest tendency for formation of this thin film among three tested RHP ensembles. The solution is stirred to tear the thin film into suspended flakes for visualization.

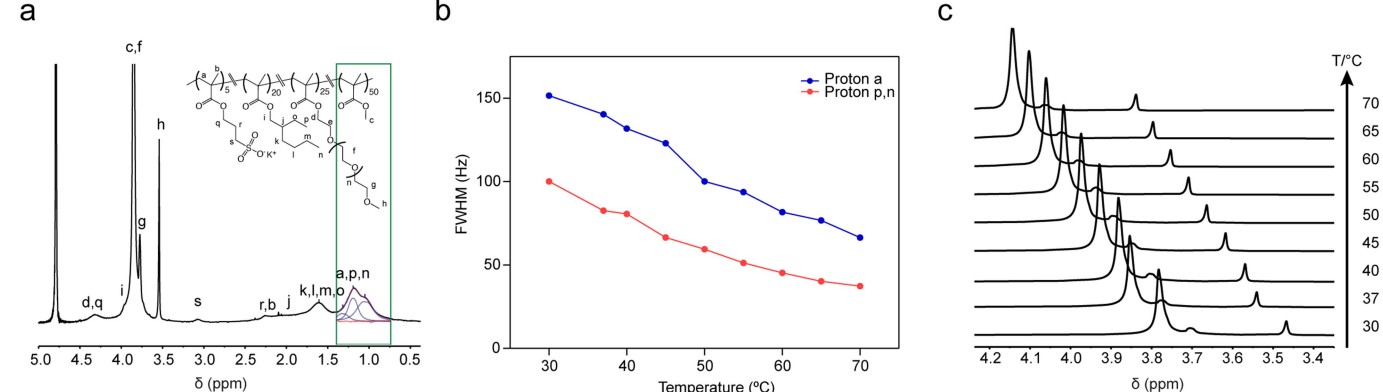

**Extended Data Fig. 3 | Temperature-dependent $^1$H-NMR spectrum of RHP4 in D$_2$O.** (a) $^1$H-NMR spectrum of RHP4 in D$_2$O at 37 °C and its chemical structure. (b) The FWHM of proton peaks (a, p, and n) as a function of temperature (30–70 °C). (c) Part of $^1$H-NMR spectra of RHP4 as a function of temperature (30–70 °C) including proton peaks of the OEGMA side chain.

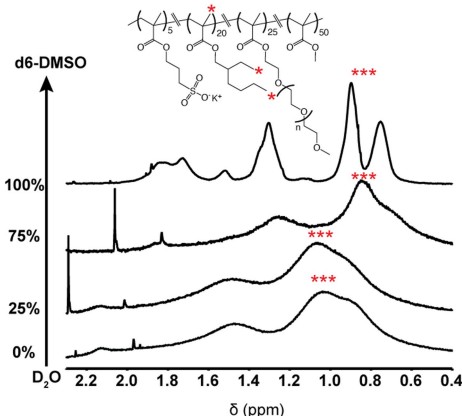

**Extended Data Fig. 4 | The effect of solvent polarity on ¹H-NMR spectrum of RHP4 in d⁶-DMSO/D₂O cosolvent.**

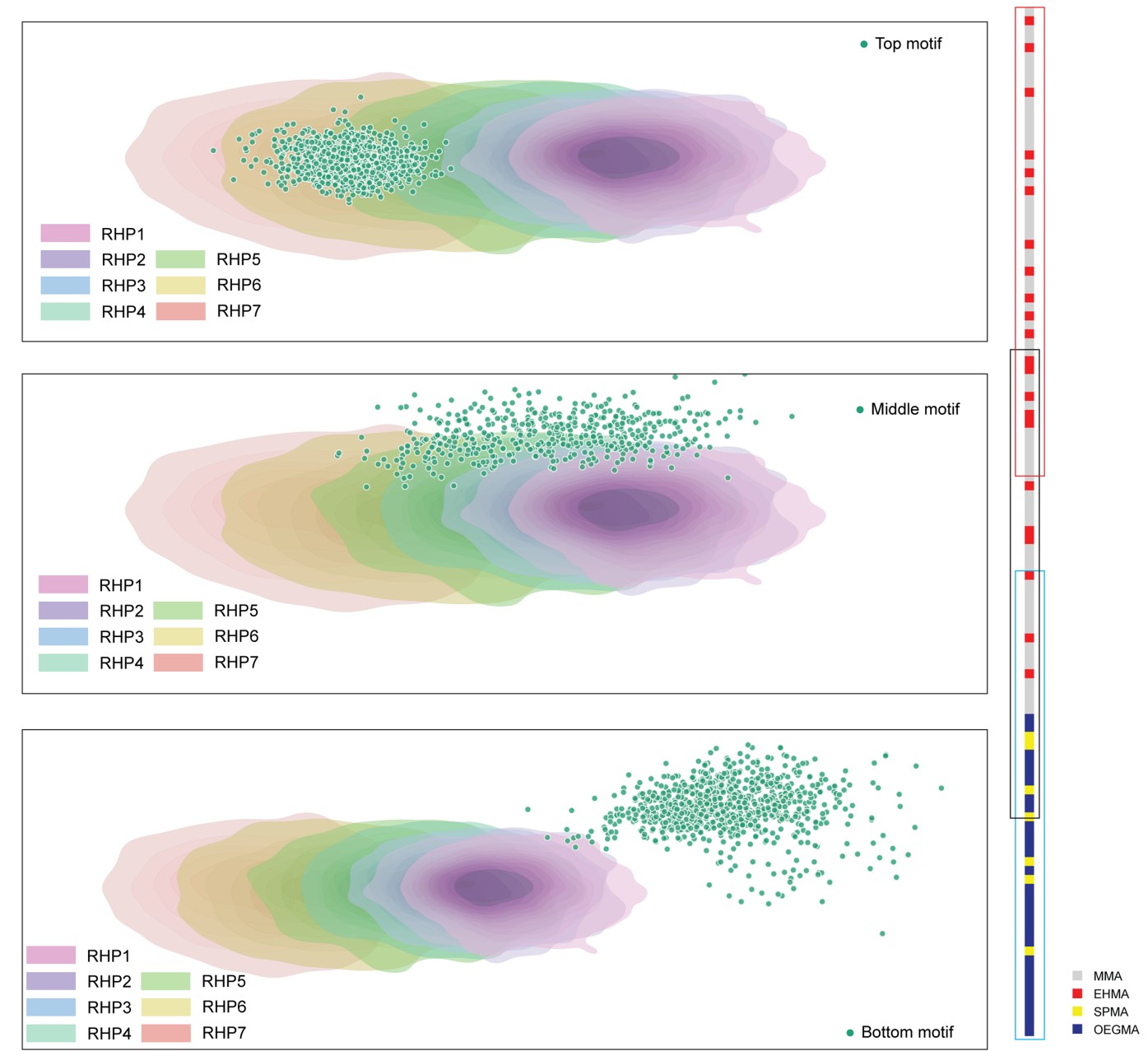

**Extended Data Fig. 5 | The distribution of DHP2 segments in PCA space.** From top to bottom, the distribution of segments from hydrophobic region, amphiphilic region, and hydrophilic region along DHP2 chains were shown.

a

b

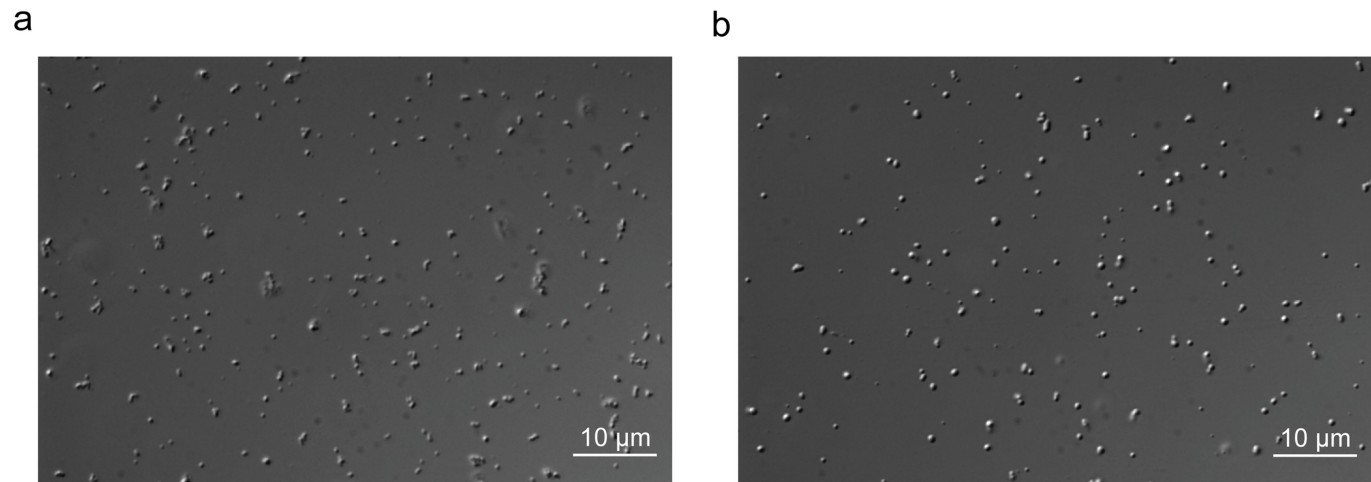

**Extended Data Fig. 6 | Differential interference contrast (DIC) images of (a) RHP8 (b) RHP9 phase-separated droplets.** Each sample is 1 mg/ml in sodium phosphate buffer (50 mM, pH 7.0).

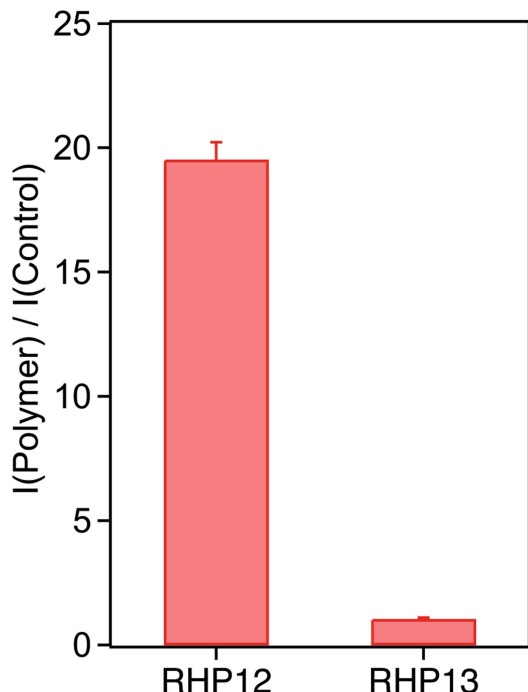

**Extended Data Fig. 7 | Folding status of AqpZ-eGFP in the presence of 0.2 wt% RHPs based on the eGFP fluorescence.** Error bar is 1 s.d and n = 3.

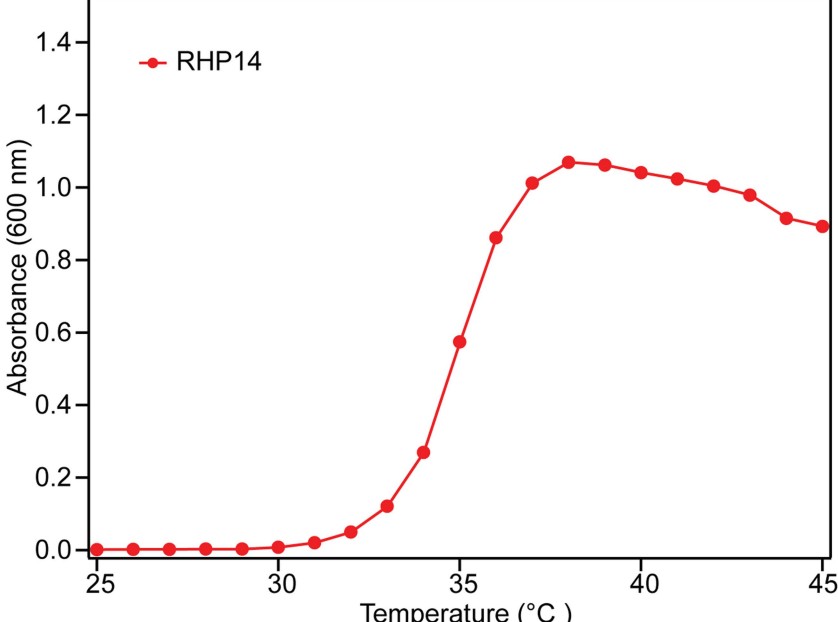

**Extended Data Fig. 8 | Temperature-dependent turbidimetry for RHP14 ensemble.** The polymer solution is 1 mg/ml in sodium phosphate buffer (50 mM, pH 7.0).

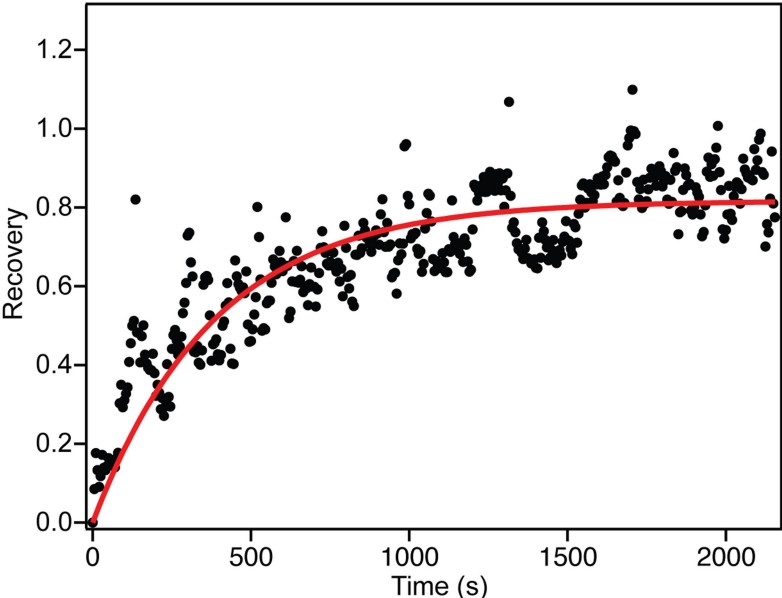

**Extended Data Fig. 9 | FRAP analysis of liquid-like coacervates made from RHP10 ensemble.** The recovery trace shows the normalized recovery of a bleached region. Solid red curve fits to an exponential function (see FRAP method section).

**Extended Data Table 1 | The conversion from amino acids to RHP monomers (_n_ = 4)**

| ID | Hydrophobicity | Amino acid | Monomer equivalent |
|----|----------------|------------|--------------------|
| 1 | Hydrophobic | Cys, Tyr, Ala, Thr, and Gly | MMA |
| 2 | Hydrophilic | Ser, Gln, His, Asn, and Pro | OEGMA |
| 3 | Very hydrophobic | Leu, Ile, Phe, Trp, Val, and Met | EHMA |
| 4 | Charged | Glu, Asp, Arg, and Lys | SPMA or DMAEMA |

