## [Peer Review File · Nature]

Manuscript Title: Population-Based Heteropolymer Design to Mimic Protein Mixtures

Reviewer Comments & Author Rebuttals

Reviewer Reports on the Initial Version:

Referees' comments:

Referee #1 (Remarks to the Author):

Xu and co-workers use statistically designed heteropolymer sequences to make protein mixtures that can mimic the function of biological fluids. Xu Lab has previously made seminal contributions by using heteropolymers to show the preservation of protein function (Ref. 17), proton transport (Ref. 22), and nano-dispersed enzymes (10.1038/s41586-021-03408-3). The critical question is how this study is different or a significant extension of the previous work to warrant publication in Nature. As I outline below, there is no doubt that a considerable effort went into this paper, and many interesting insights were obtained from these efforts.

The first part of the paper deals with defining segmental level characteristics of globular and membrane proteins using PCA. I am confused as to why such a complicated representation in terms of PC1 and PC2 is needed if the input features are based on the hydrophobicity of the amino acids in only 4 categories. Moreover, as PC2 hardly distinguishes the two types of proteins, why not just use PC1 as the sole order parameter? The essential missing part is the physical intuition, as it is challenging to follow the discussion associated with this part. Also, it is unclear how the features identified in previous work, especially Ref. 17, differ in superiority from this new analysis.

The conclusion from the single-chain pulling studies is summarized as: "Together, these single-chain studies show that an RHP ensemble contains subpopulations but has a defined range of segmental hydrophobicity similar to mixtures of proteins." This relatively broad and vague statement is followed by a hypothesis regarding recapitulating protein behaviors in biological fluids. Similarly, a tacit assumption is made that the biological fluids can be modeled as a mixture of proteins or polymers; therefore, other macromolecules and solvent additives don't matter. It is hard to believe such an assumption is valid even if the polymer mixtures can provide some interesting properties.

The following section deals with testing the membrane protein folding with or without RHPs. If I understand correctly, the control case is the cytosol solution itself, which is not very good with the protein yield, but certain RHPs are better in providing a higher protein folding. Are the authors proposing that biological fluids cannot support proteins to navigate "random encounters?" The results presented in Figure 2 are still helpful for designing synthetic fluids to promote protein folding. I believe these results overlap with previously published results in Ref. 17. Also, I find the lack of other controls, such as adding common crowding agents such as Ficoll, PEG, etc., to complement the RHP results problematic. As the results can vary from significant change (30-fold) to minor change (1.1-fold) due to RHP addition, more discussion and work are needed to help understand the underlying phenomenon.

Next, the authors show higher thermostability of proteins with RHP ensembles with maximal PCA overlap and the liquid-liquid phase separation of RHP ensembles. These observations, though interesting, lack fundamental hypotheses and the ability to build upon these results. At this point, it is hardly surprising that a protein or a polymer can undergo phase separation at the given conditions, as numerous previous studies have shown this to occur in many systems. What do we learn from this about LLP?

The authors present molecular simulation results to connect the experimental observations with molecular details, but in a different context than most laboratory experiments. I had great difficulty following these results beyond the simulated observations and how these help explain the function of RHP ensembles.

Overall, I felt that the paper could benefit from a more extended format where the results are presented with clarity to a non-expert, which is likely for every reader, given the breadth of topics covered. This will also allow the authors to look at the results in a more balanced way and highlight issues that are not understood rather than simply making it sound like a testable scientific justification exists. I understand the logic of keeping it short for a short-term higher impact, but then the authors should make more effort to explain their results and simplify the text to remove technical jargon unless necessary.

Referee #2 (Remarks to the Author):

This manuscript presents an excellent and detailed analysis of synthetic random heteropolymers in analogy to proteins. While not focusing on sequence specific effects, the authors perform an analysis at the compositional and segmental level, demonstrating the ability of polymers within the ensemble to behave as disordered, partially folded, and folded globular molecules. Furthermore, the authors demonstrate strong synergy between random heteropolymers and proteins as a function of their composition, as described by a principle components analysis. Polymers that were more similar to the protein of interest showed an enhanced ability to stabilize folding during translation, as well as provide protection against thermal denaturation. Lastly, the ability to provide protection against denaturation was enhanced by the ability of a polymer to undergo liquid-liquid phase separation, in analogy to membraneless organelles. This work builds off of previous, more focused efforts by the same group, and does an excellent job combining computational, analytical, and experimental results. The manuscript is well written and the data presented in an accurate and meaningful way with appropriate treatment of uncertainties. The breadth of the results demonstrating analogies with biological systems is very exciting and would be of interest to a diverse audience, and I strongly support its publication.

In terms of suggested improvements, I noted that author contributions section did not include a description for Haotian Chen, Tao Jiang, Alfredo Alexander-Katz, Carlos Bustamante, or Haiyan Huang. Please address.

Referee #4 (Remarks to the Author):

Xu and coworkers report a tour-de-force effort to design, simulate, synthesize, and characterize ensembles of random heteropolymers (RHPs) that mimic protein-rich fluid environments such as the cytosol and blood sera. RHP ensembles are generated with varying degrees of hydrophilicity, hydrophobicity, and charge. Principal component analysis is used to classify RHP ensembles and to compare RHP ensembles to naturally occurring globular and membrane proteins. RHP proteins are evaluated using myriad approaches to understand their physical properties and performance as synthetic polymeric replacements for protein-rich fluids.

This work represents an original and significant advance from earlier contributions by Alexander-Katz and Xu exploring the fundamental physics and application of RHPs. In this work, the authors embrace biomimicry by using RHP solutions—which are completely synthetic materials—to enhance biological activity in challenging scenarios such as the expression of membrane proteins and the protection of proteins against thermal denaturation. The demonstration of liquid–liquid phase separation (LLPS) is less novel given both historic and recent interests in complex coacervation (see: <https://pubs.acs.org/doi/pdf/10.1021/ma60059a040>, <https://pubs.rsc.org/en/content/articlelanding/2020/sm/d0sm00001a>). The overall work still stands out as impressive, interdisciplinary, and interesting to broad audiences. I recommend publication after the authors address the following technical points and clarifications:

1. Line 74: I was unfamiliar with the terminology “latent embedding” and encourage use of a more general descriptor such as “envelope”
2. Following PCA, the authors have connected PC1 to hydrophobicity. Could the authors comment on any physicochemical features related to PC2?
3. Line 83: How is a “segment” defined? Does a segment represent a single monomer that has been re-assigned from an amino acid to an RHP monomer, or does it refer to the local environment produced by several monomers? Moreover, the authors use several different phrases to refer to “segmental hydrophobicity”; in line 83, the phrase “segmental distribution” might be clearer as “distribution of segmental hydrophobicity” or “segmental hydrophobicity distribution” – if that is the intended use. As written, the definition of a “segment” is vague.
4. Line 105: Could the authors elaborate what they mean by “results from 30 and 97 chains” for SMFS experiments? Were these experiments run during separate sessions?
5. Line 120: Could the authors verify whether the standard deviation is reported to an appropriate number of significant digits?
6. The authors have generated an extensive SI file; however, there are very few single-molecule force trajectories shown in this report. It would be helpful for the authors to include supplementary plots illustrating all of the SMFS trajectories (perhaps sorted by classification).
7. Line 145: the authors mention a critical overlap concentration; could the authors quantify this

value in the main text?

8. Fig. 2: Is OmpT also generally more or less challenging to express than the other proteins of interest? The relative yield with RHPs does not seem as enhanced as for OmpT than for PepTso or AqpZ.

9. Fig. 2F: I find it surprising that RHP7 has slightly improved performance over RHP6 despite less similarity of RHP7 with OmpT. Could the authors comment on this discrepancy?

10. Fig. 3: The “No FBS”, “Native FBS”, and “Heated FBS” conditions are inconsistent with the main text discussion and remain unclear from the caption. Which test condition(s) contain RHP?

11. Lines 187–202: The authors’ discussion of LLPS is not technically accurate. LLPS can emerge as a bulk (non-microscopic) phenomenon; for example, liquid–liquid extraction due to the immiscibility of aqueous and organic phases is a large-scale application of LLPS. I encourage the authors to clarify that they are referring to microphase separation and/or coacervation, which can drive the formation of membraneless organelles in cellular systems. This discussion also requires copy-editing to meet the standards of technical publishing, including both a grammatical review and a closer look at the references. I noticed 2 separate sets of references in the main document, which appears to be the source of incorrect numbering throughout the LLPS discussion.

12. RHPs 8–12 should be compositionally defined in a supplementary table in similar fashion to RHPs 1–7.

13. Line 209/Fig. S11: please include the equation used to fit the red line.

14. Line 222/Fig. S14: It would be helpful to include RHP1 in this figure.

15. Lines 225–231: I do not understand the argument that “the right segments show up at the right place and right time.” Could the authors expand upon the current explanation?

16. Line 233: The authors note that RHPs form compact globules; however, a diameter of 8.8 nm seems on the large side for a 36 kDa biomolecule (if I recall correctly, proteins of this molecular weight would typically span 4–5 nm in diameter). I suggest the authors compare their SAXS data with scaling arguments for folded/ denatured proteins: <https://pubs.acs.org/doi/10.1021/bi991765q>

17. Line 235: Visual snapshots from the MD simulation would support the arguments about RHP conformation and surface exposed vs. buried segments.

18. Fig. S20: I don’t understand how these data suggest unraveling – could the authors elaborate upon this interpretation? I encourage including this data side-by-side or overlaid with the SAXS data in Fig. 5A.

19. The manuscript contains several typos and would benefit from minor corrections including but not limited to:

- Fig. 3B: "MTT Assay" (caption) should be defined.
- Fig. 3C: "Proteinase K" is misspelled.
- Table S3: Monomers 5 and 6 are both OEGMA ($M_n=300$ Da); I suspect one of these monomers should be $M_n=500$ Da.
- Fig. 4A: I recommend labeling the sub-sub-panels (e.g., "Fig. 4A bottom left") as separate panels.
- Line 224: "Membraneless compartment" is misspelled.
- Fig. S15: The yellow lines in the middle graph are difficult to read; is there a technical reason for the different colors used in the left and center plots?
- Fig. 5D: "HLB" (y-axis) should be defined.

Author Rebuttals to Initial Comments:

Reviewer #1:

1. *Xu and co-workers use statistically designed heteropolymer sequences to make protein mixtures that can mimic the function of biological fluids. Xu Lab has previously made seminal contributions by using heteropolymers to show the preservation of protein function (Ref. 17), proton transport (Ref. 22), and nano-dispersed enzymes (10.1038/s41586-021-03408-3). The critical question is how this study is different or a significant extension of the previous work to warrant publication in Nature.*

Author response: We thank the reviewer for taking the time and helping us to improve the manuscript. We fully agree on the importance of differentiating from prior work, which we did not do sufficiently in the initial submission. We also appreciated the reviewer's suggestion to expand the text to provide readers with materials to better digest the information. We modified the manuscript accordingly as detailed below.

Connections to previous work Studies in Ref. 17 (*Science*, 359, 1239, 2018.), Ref. 20 (*Nature*, 577, 216, 2020.), and this manuscript built upon each other and reflected how we matured in translating protein sequence information to synthetic materials. Studies in Ref. 17 were inspired by protein's surface patchiness and guided by the block length distribution in soluble proteins. Studies in Ref. 20 represent a critical transition in our pursuit of RHPs as protein mimics, given that they highlighted the importance of flanking segments in RHPs and demonstrated RHPs can be designed beyond mimicking intrinsically disordered proteins. More importantly, they led us to consider inherent heterogeneity within each RHP ensemble.

1-D sequence analysis lacks segment arrangement information Previous studies mainly focused on the distribution of segment/block length. The segment/block length distribution is a one-dimensional piece of sequence information. Experimentally, the 1-D sequence information has been shown effective where the RHP design focused on a single function under specific conditions, e.g., being anchored at polar/nonpolar interfaces for enzyme stabilization under non- biological conditions or inserted into lipid bilayer for proton transport. In this contribution, we aimed to design multi-functional RHPs both as a population and as a subset to mimic protein mixtures in biological fluids. This new requirement is substantially more challenging to encompass all possible random encounters, including those with low probabilities, to ensure a predictable outcome.

2-D sequence analysis is effective in guiding RHP design for simultaneous modulation of collective and individual behaviors of RHPs. Using the reviewers' insights as a basis, we performed further studies with new RHPs designed based on proteins undergoing LLPS to substantiate these claims. Besides modulating the cloud point temperatures of RHP ensembles to be biologically relevant, we evaluated how RHPs affect protein folding and stability with and without LLPS. We also evaluated the competition between DNA-RHP interactions vs. DNA pairing within the liquid droplets, an important factor to regulate DNA transcription. The presence of RHPs in the liquid droplets did not compromise stability of double-stranded DNA. When two RHP droplets containing complementary single-stranded DNA fuse, the association of DNA proceeds. Together, these results confirmed the multi-functional nature of designed

RHPs at different length scales. This multi-functional nature of RHPs highlights the key design rule to translate protein sequence information to synthetic materials and opens many possibilities to engineer synthetic living systems.

2. *As I outline below, there is no doubt that a considerable effort went into this paper, and many interesting insights were obtained from these efforts.*

Author response: Thank you very much.

3. *I am confused as to why such a complicated representation in terms of PC1 and PC2 is needed if the input features are based on the hydrophobicity of the amino acids in only 4 categories. Moreover, as PC2 hardly distinguishes the two types of proteins, why not just use PC1 as the sole order parameter? The essential missing part is the physical intuition, as it is challenging to follow the discussion associated with this part.*

Author response: Thank you for pointing this out. We fully agree with the reviewer’s assessment. We have revised the manuscript adding following details regarding the 2-D sequence analysis. The additional information substantially improved the clarity and technical quality of the revised manuscript. We are very grateful.

The sequential arrangement of segments along an RHP chain affects inter-segment interactions, the effective chemical characteristics of each segment, and the interchain interactions. Thus, we expanded the protein sequence analysis into a second dimension, where the segment sequence information is mainly carried in PC2 but is encoded in both PC1 and PC2. This is illustrated in a 2-D PCA map of a library of hypothetical polymer chains containing four segments with different segment sequences (Figure R1). These chains are indistinguishable based on 1-D analysis described in Ref. 17. Here, the 2-D sequence analysis can clearly capture the differences in their segment sequences. The dependence of PC1 on the segment sequence, which can be approximated as the apparent hydrophobicity of each chain, is also evident. These sequence analyses are consistent with results using block copolymer analogs of RHPs. When protein sequences are mapped into the same PCA space, the PC2 distribution of proteins can be described by a normal distribution centered close to the PC2 value of the sequence 12 below. This indicates that most proteins tend to have alternating hydrophobic/hydrophilic segments along the chain (like sequence 12) and avoid segregating hydrophobic blocks into long blocks (such as sequence 1 or sequence 24). Translating this information into the RHP design represents a step toward capturing protein’s sequence complexity.

Figure R1 (Left) 24 permuted sequences with RHP4 composition in descending order of PC2 value from top to bottom. (Right) The projection of 24 permuted sequences and RHP4 ensemble onto the PCA space.

The fact that PC2 hardly distinguishes the two types of proteins suggests that protein sequences from either class arrange neighboring segments in a similar manner. We hypothesized that a biological system purposely evolves in such way as to avoid long hydrophobic blocks to modulate inter-chain interactions. RHP ensembles overlap with proteins well in PC2 dimension. However, the control polymer ensemble (DHP2) shows extreme PC2 values (Figure R2). This is because DHP arrange neighboring segments in a blocky fashion (segregating hydrophobic monomers into a single block). As a result, DHP2 deviates from the protein sequence drastically and has quite different phase behavior. DHP2 was also unable to mediate protein folding. Together, we believe demonstrating PC2 and its underlying physical details is important for understanding the complexity of a biological system and designing future materials.

Figure R2 Comparison of RHPs and DHP on the 2-D PCA map. The start/middle/end motif accounts for 50-mers from the PMMA-co-EHMA block, in-between block, or POEGMA-co- SPMA block, respectively.

4. Also, it is unclear how the features identified in previous work, especially Ref. 17, differ in superiority from this new analysis.

Author response: We thank the reviewer for pointing this out and fully agree that the previous version of the manuscript did not make efforts to delineate the difference from previous and highlight the new advances. We agree with the reviewer that the lack of this critical information significantly weakens the manuscript and have thus since revised the manuscript by highlighting the inherent differences which we believe represent a significant step forward in the rational design of synthetic polymers to recapitulate protein's phase behavior.

Experimentally, the RHP results have been very robust for two reasons. First, the target readouts are based on the scenarios where RHPs are pinned at either interfaces or inserted into a lipid bilayer that limits the RHP chain conformation. Only a limited subset of inter-segment interactions can be sampled. Secondly, studies in Ref. 17 were based on a combination of analysis of protein's block length distribution, experimental screening, and scientific intuition.

The 1-D sequence analysis based on block length distribution has been effective and can be viewed as a special case for the 2-D analysis. When 2-D sequence analysis results from one RHP ensemble (RHP4) are superimposed in Figure R1, they satisfy the design criteria based on 2-D sequence analysis and occupy a narrower PC2 space than that occupied by hypothetical tetra-block polymers.

In the supporting information, we added 1-D sequence analysis results. In Ref. 17, the protein sequence was mapped into a histogram of block length distribution where a block is a consecutive segment of single amino acids (either hydrophobic or hydrophilic). Such 1-D analysis is sufficient for effectively realizing protein stabilization in nonaqueous media. However, as we increase the system complexity by considering the whole population of the proteins randomly encountering the whole population of the RHP chains, the design based on block length is no longer effective. There is no critical threshold of hydrophobic block length to differentiate globular proteins and membrane proteins (Fig. S1). When designing RHPs to mimic such a broad distribution of the hydrophobic blocks in proteins (Fig. S2), we cannot differentiate which RHP can achieve better protein stabilization in aqueous environments as each RHP has the same distribution based on the previous analysis.

The 2-D sequence analysis not only captures interactions within a fixed short-ranged distance, but also in a solution when both proteins and polymers are randomly distributed. Experimental results demonstrated a predictable outcome regarding the way in which RHPs interact with single protein and the way in which an RHP ensemble interacts with itself or with a collection of proteins, e.g. in the study of LLPS and FBS stabilization, respectively.

5. *The conclusion from the single-chain pulling studies is summarized as: Together, these single-chain studies show that an RHP ensemble contains subpopulations but has a defined range of segmental hydrophobicity similar to mixtures of proteins." This relatively broad and vague statement is followed by a hypothesis regarding recapitulating protein behaviors in biological fluids. Similarly, a tactic assumption is made that the biological fluids can be modeled as a mixture of proteins or polymers; therefore, other macromolecules and solvent additives don't matter. It is hard to believe such an assumption is valid even if the polymer mixtures can provide some interesting properties.*

Author response: We thank the reviewer for pointing this out and agree with the reviewer's assessment. We have revised the statement as "Together, an RHP ensemble that matches the PCA space of protein mixtures has a defined range of segmental characteristics and can mimic the proteins' conformational diversity."

Experimentally, we chose FBS because it contains 1000+ components including proteins and other molecules and can be treated as a miniature model of biological fluids. We modified the manuscript by highlighting that RHPs can be designed to capture the optimal interaction range occurring in biological fluids, as shown experimentally by the result that RHP can enhance FBS thermal stability strongly. The statement reads as "The overlapping PCA regions between proteins and each RHP ensemble indicate similarity in their segmental chemical characteristics, which define the range of interactions during random encounters in biological fluids. We hypothesize that to design RHPs as protein mixtures, it is more essential to capture

the range of intra- and intermolecular interactions within biological fluids rather than replicating their exact compositions, which fluctuate and remain undefined.”.

- 6. The following section deals with testing the membrane protein folding with or without RHPs. If I understand correctly, the control case is the cytosol solution itself, which is not very good with the protein yield, but certain RHPs are better in providing a higher protein folding. Are the authors proposing that biological fluids cannot support proteins to navigate "random encounters?"*

Author response: Thanks for pointing this out. The control case is not the cytosol. The solution is a coupled cell-free transcription/translation system reconstituted from purified components necessary for *E.coli* translation. Thus, unnecessary components, including many proteins in the cytosol, were removed. This makes the control case more relevant and accurate as we rule out potential stabilization effects from real protein mixtures. We revised the manuscript to avoid potential confusion.

- 7. The results presented in Figure 2 are still helpful for designing synthetic fluids to promote protein folding. I believe these results overlap with previously published results in Ref. 17.*

Author response: Thank you for pointing this out. Cell-free expression is an ideal assay to probe how proteins interact with their surrounding media. It was used here to interrogate whether RHP ensembles can mimic certain functions of the protein mixtures, i.e. facilitating protein folding during structural fluctuation.

We fully appreciate and agree with the reviewer’s impression about overlapping. We modified the manuscript accordingly to highlight significant advances (enhanced protein translation efficiency from the biomachinery, such as ribosome, as well as correlation between protein folding and RHP/protein PCA overlap) and made the text more concise.

- 8. Also, I find the lack of other controls, such as adding common crowding agents such as Ficoll, PEG, etc., to complement the RHP results problematic.*

Author response: Thank you for this comment. We agree with the reviewer that crowding agents can stabilize proteins attributing to the macromolecular crowding effect. Thus, we purposely kept the RHP concentration below the critical overlapping concentration to minimize any crowding effect. Controlled experiments confirmed that PEG-8000, amphipol, and DOPC liposomes were not effective in facilitating protein folding at the same level as RHPs. In fact, Amphipol, a short amphipathic polymer that is used to solubilize membrane proteins in aqueous solutions, showed an inhibitory effect on protein expression.

- 9. As the results can vary from significant change (30-fold) to minor change (1.1-fold) due to RHP addition, more discussion and work are needed to help understand the underlying phenomenon.*

Author response: Thank you for bringing this to our attention. The metric we used to measure the effect of RHP on assisting protein folding was the ratio of eGFP fluorescence intensity with

RHP to that without RHP at the end of expression. OmpT can partially fold even without RHP as shown in Fig. S14, unlike the other two proteins. This is consistent with its PC1 values for these proteins. Thus, there is a relatively high intensity in the absence of RHP and seemingly less significant enhancement by adding RHPs.

To elaborate on this performance discrepancy, we changed the sentence in the manuscript as, “OmpT has a fairly good folding status without RHPs (Fig. S14); this was attributed to its lower apparent hydrophobicity compared to the other MPs, as seen by its higher PC1 value. RHP1-3 with higher PC1 values were the most effective in mediating OmpT folding, whereas RHP 6-7 with lower PC1 values have deleterious effects.”.

10. Next, the authors show higher thermostability of proteins with RHP ensembles with maximal PCA overlap and the liquid-liquid phase separation of RHP ensembles. These observations, though interesting, lack fundamental hypotheses and the ability to build upon these results. At this point, it is hardly surprising that a protein or a polymer can undergo phase separation at the given conditions, as numerous previous studies have shown this to occur in many systems. What do we learn from this about LLP?

Author response: We thank the reviewer for his/her inputs and appreciate the opportunity to modify the manuscript with better delineation. We fully agree with the reviewer that LLPS has been already observed in various polymer solutions and has been a very hot topic for different communities. These are part of the reasons why we chose the topic. We believe studying LLPS in RHP ensembles addressed several critical factors that have been essentially impossible up to date, such as the system complexity, and formulation uncertainties etc. For example, as pointed out in this paper (“Considerations and Challenges in Studying Liquid-Liquid Phase Separation and Biomolecular Condensates”(https://doi.org/10.1016/j.cell.2018.12.035)), one key question to study biologically relevant LLPS is how the heuristics established from simple one or two- component systems can be transferred to more complex mixtures in cells where heterotypical interactions between proteins frequently occur. We believe that implementing species diversity and interaction heterogeneity in a designable system are the key uniqueness and advantages of RHPs. This insight distinguishes our work substantially from previous studies that mainly focus on individual molecules with known identity or blends with precise compositions, in which a single mutation can lead to dramatic changes in phase behaviors.

RHPs also capture chemical diversity in the types of intermolecular interactions, more closely echoing those experienced by proteins. As described in the revised manuscript, we performed more experiments. As shown in Fig. 4F, two complementary single-stranded DNA (ssDNA) can each be encapsulated into RHP droplets. When two RHP droplets carrying different ssDNA fuse together, the two ssDNA complex to form dsDNA. This result indicates that RHPs are compatible with biomolecules and do not disrupt their biological processes.

11. The authors present molecular simulation results to connect the experimental observations with molecular details, but in a different context than most laboratory experiments. I had great difficulty following these results beyond the simulated observations and how these help explain the function of RHP ensembles.

Author response: We fully agree with the reviewer and have made modifications to make the connection clear and more focused. MD simulation was performed to study how RHP's segments would respond to local environmental change at the nanoscopic scale. While the simulated system is in a different context, it helps to consolidate our hypothesis that an RHP chain can effectively modulate its side-chain distribution based on local environmental change. The current MD simulation serves as a complementary analysis to reduce the level of speculation in the manuscript, and we hope this may prompt the community to find more optimized approaches to probe the phase behavior of RHPs at the segmental level in the presence of external stimuli.

12. Overall, I felt that the paper could benefit from a more extended format where the results are presented with clarity to a non-expert, which is likely for every reader, given the breadth of topics covered. This will also allow the authors to look at the results in a more balanced way and highlight issues that are not understood rather than simply making it sound like a testable scientific justification exists. I understand the logic of keeping it short for a short-term higher impact, but then the authors should make more effort to explain their results and simplify the text to remove technical jargon unless necessary.

Author response: We agreed with the reviewer and modified the manuscript accordingly. Based on the suggestions from both reviewers and the editorial office, we restructured the manuscript and performed additional experiments. We believe the revised manuscript better captures our key message and hope the reviewer may find the revised version to be more balanced with improved clarification.

Reviewer #2:

This manuscript presents an excellent and detailed analysis of synthetic random heteropolymers in analogy to proteins. While not focusing on sequence specific effects, the authors perform an analysis at the compositional and segmental level, demonstrating the ability of polymers within the ensemble to behave as disordered, partially folded, and folded globular molecules. Furthermore, the authors demonstrate strong synergy between random heteropolymers and proteins as a function of their composition, as described by a principle components analysis. Polymers that were more similar to the protein of interest showed an enhanced ability to stabilize folding during translation, as well as provide protection against thermal denaturation. Lastly, the ability to provide protection against denaturation was enhanced by the ability of a polymer to undergo liquid-liquid phase separation, in analogy to membraneless organelles. This work builds off of previous, more focused efforts by the same group, and does an excellent job combining computational, analytical, and experimental results. The manuscript is well written and the data presented in an accurate and meaningful way with appropriate treatment of uncertainties. The breadth of the results demonstrating analogies with biological systems is very exciting and would be of interest to a diverse audience, and I strongly support its publication.

Author response: We thank the reviewer for his/her inputs.

1. author contributions section did not include a description for Haotian Chen, Tao Jiang, Alfredo Alexander-Katz, Carlos Bustamante, or Haiyan Huang. Please address.

Author response: Thank you for pointing this out. We have included a description for these authors in the author contribution section.

Reviewer #3:

Xu and coworkers report a tour-de-force effort to design, simulate, synthesize, and characterize ensembles of random heteropolymers (RHPs) that mimic protein-rich fluid environments such as the cytosol and blood sera. RHP ensembles are generated with varying degrees of hydrophilicity, hydrophobicity, and charge. Principal component analysis is used to classify RHP ensembles and to compare RHP ensembles to naturally occurring globular and membrane proteins. RHP proteins are evaluated using myriad approaches to understand their physical properties and performance as synthetic polymeric replacements for protein-rich fluids.

This work represents an original and significant advance from earlier contributions by Alexander-Katz and Xu exploring the fundamental physics and application of RHPs. In this work, the authors embrace biomimicry by using RHP solutions—which are completely synthetic materials—to enhance biological activity in challenging scenarios such as the expression of membrane proteins and the protection of proteins against thermal denaturation. The demonstration of liquid–liquid phase separation (LLPS) is less novel given both historic and recent interests in complex coacervation (see: <https://pubs.acs.org/doi/pdf/10.1021/ma60059a040>, <https://pubs.rsc.org/en/content/articlelanding/2020/sm/d0sm00001a>). The overall work still stands out as impressive, interdisciplinary, and interesting to broad audiences. I recommend publication after the authors address the following technical points and clarifications.

Author response: We thank the reviewer for their perspective on our work.

- 1. The demonstration of liquid–liquid phase separation (LLPS) is less novel given both historic and recent interests in complex coacervation*

Author response: We thank the reviewer for his/her inputs and appreciate the opportunity to modify the manuscript with better delineation. We fully agree with the reviewer that LLPS has been already observed in various polymer solutions and has been a very hot topic for different communities. These are part of the reasons why we chose the topic. We believe studying LLPS in RHP ensembles addressed several critical factors that have been essentially impossible up to date, such as the system complexity, and formulation uncertainties etc. For example, as pointed out in this paper (“Considerations and Challenges in Studying Liquid-Liquid Phase Separation and Biomolecular Condensates”(<https://doi.org/10.1016/j.cell.2018.12.035>)), one key question to study biologically relevant LLPS is how the heuristics established from simple one or two- component systems can be transferred to more complex mixtures in cells where heterotypical interactions between proteins frequently occur. We believe that implementing species diversity and interaction heterogeneity in a designable system are the key uniqueness and advantages of RHPs. This insight distinguishes our work substantially from previous studies that mainly focus on individual molecules with known identity or blends with precise compositions, in which a single mutation can lead to dramatic changes in phase behaviors.

RHPs also capture chemical diversity in the types of intermolecular interactions, more closely echoing those experienced by proteins. As described in the revised manuscript, we performed more experiments. As shown in Fig. 4F, two complementary single-stranded DNA (ssDNA) can each be encapsulated into RHP droplets. When two RHP droplets carrying

different ssDNA fuse together, the two ssDNA complex to form dsDNA. This result indicates that RHPs are compatible with biomolecules and do not disrupt their biological processes.

2. Line 74: I was unfamiliar with the terminology “latent embedding” and encourage use of a more general descriptor such as “envelope”

Author response: Thank you for bringing this to our attention. In machine learning, latent embedding is equivalent to low-dimensional representation of data. To present this more clearly in the manuscript, we have replaced the term “latent embedding” with “low-dimensional vector”.

3. Following PCA, the authors have connected PC1 to hydrophobicity. Could the authors comment on any physicochemical features related to PC2?

Author response: Thank you for pointing this out. PC2 correlates with the sequential arrangement of different blocks within the 50-mer. To clearly illustrate the physical details of two-dimensional representation, we manually constructed a set of tetra-block polymers with RHP4’s composition, as shown below. By permutating the four blocks (a consecutive motif of single monomer) as shown below, the PC2 value decreased from top sequence to bottom sequence. This suggests that PC2 dimension correlates with the neighboring blocks at the sequence level. Thus, it allows us to investigate short-ranged interactions between neighboring segments, as embedded in the protein sequences. In addition, we would like to highlight that the change of neighboring segments will change the apparent hydrophobicity of the chain, as reflected in the change of the PC1 value. When protein sequences were mapped into the same PCA space, we noticed that PC2 distribution of proteins could be described by a normal distribution centered close to the PC2 value of the sequence 12 below. This indicates that most proteins tend to have alternating hydrophobic/hydrophilic segments along their sequences (similar to sequence 12) and avoid segregating hydrophobic blocks into long blocks (such as sequence 1 or sequence 24). We believe that nature purposely evolved in this way and that transforming this information into the design of synthetic polymers represents a step toward recapitulating protein’s sequence complexity.

Figure R1 (Left) 24 permutated sequences with RHP4 composition in descending order of PC2 value from top to bottom. (Right) The projection of 24 permutated sequences and RHP4 ensemble onto the PCA space.

4. Line 83: How is a “segment” defined? Does a segment represent a single monomer that has been re-assigned from an amino acid to an RHP monomer, or does it refer to the local environment produced by several monomers? Moreover, the authors use several different

phrases to refer to “segmental hydrophobicity”; in line 83, the phrase “segmental distribution” might be clearer as “distribution of segmental hydrophobicity” or “segmental hydrophobicity distribution” – if that is the intended use. As written, the definition of a “segment” is vague.

Author response: We thank the reviewer for pointing this out. We agree that the definitions of block, segment and 50-mers were confusing in the initial submission. In the revised manuscript, we made following modifications: We specifically defined “block” as the consecutive residues of the same hydrophobicity on a binary assignment. This is consistent with previous reports. We clarified the selection of 50-mer for the 2-D PCA analysis. We noted that 50-mer selection is arbitrary and based on previous studies in peptide-polymer conjugates. We purposely reserved the segment to be loosely defined in the spirit of the equivalent freely jointed chain model when we probed RHP-protein interactions. We believe this choice is more meaningful technically as we need to consider the chain dynamics of both RHP and proteins and their tendency to locally modulate their conformations based on their surroundings. In addition, we have revised the manuscript and avoided unclear expressions such as “segmental distribution”.

5. *Line 105: Could the authors elaborate what they mean by “results from 30 and 97 chains” for SMFS experiments? Were these experiments run during separate sessions?*

Author response: These experiments were run during the same session using the same batch of samples. We started by collecting SMFS data for 30 chains and counted the number of chains for each class of trajectories. Subsequently, we collected SMFS data for an additional 67 chains and did the same analysis for a total of 97 chains. The distribution of trajectories gathered from 30 chains and 97 chains were similar and thus the current results from 97 chains were extrapolated to the whole population of RHPs.

For better clarification, we rephrased the sentence as, “results from the first 30 chains are statistically similar to those from a total of 97 chains” in the main text.

6. *Line 120: Could the authors verify whether the standard deviation is reported to an appropriate number of significant digits?*

Author response: Thank you for pointing this out. We have verified and updated the reported value with 29.0 ± 22.3 kcal/mol (mean \pm s.d.).

7. *The authors have generated an extensive SI file; however, there are very few single-molecule force trajectories shown in this report. It would be helpful for the authors to include supplementary plots illustrating all of the SMFS trajectories (perhaps sorted by classification).*

Author response: Thank you for pointing this out. We have added all of the SMFS trajectories to the SI (Fig.S9-S12).

8. *Line 145: the authors mention a critical overlap concentration; could the authors quantify this value in the main text?*

Author response: Thank you for pointing this out. We have added the suggested content to the manuscript as: “ Experimentally, the RHP solution concentration was set at 0.2 wt %, well below the RHP’s critical overlap concentration (>10 wt% for RHPs studied) to minimize crowding effects.”

9. *Fig. 2: Is OmpT also generally more or less challenging to express than the other proteins of interest? The relative yield with RHPs does not seem as enhanced as for OmpT than for PepTso or AqpZ.*

Author response: Thank you for pointing this out. As shown in Fig. S14, without RHP, OmpT- eGFP exhibited nontrivial fluorescence whereas AqpZ-eGFP did not. This difference suggests that OmpT is less challenging to express and fold than other membrane proteins of interest. The metric we used to quantitate the effect of RHP on assisting protein folding was the ratio of eGFP fluorescence intensity (I) with RHP to the intensity without RHP. Since OmpT can partially fold even without RHP, this leads to a relatively high I (no RHP) and seemingly less significant enhancement, by adding RHPs in OmpT than in PepTso or AqpZ.

10. *Fig. 2F: I find it surprising that RHP7 has slightly improved performance over RHP6 despite less similarity of RHP7 with OmpT. Could the authors comment on this discrepancy?*

Author response: Thank you for pointing this out. As shown in Fig. 2D, the fluorescence intensity of OmpT-eGFP decreased upon adding RHP6 or RHP7, suggesting that either RHP ensemble destabilizes OmpT during the protein expression. RHP6 and RHP7 have more hydrophobic segments than OmpT does. This mismatch of segment space between two RHPs and OmpT could lead to strong hydrophobic RHP-OmpT interactions that outcompete those interactions governing OmpT folding, and thereby destabilize OmpT. Although RHP7 is less similar to OmpT than is RHP6, and is also more hydrophobic, RHP7 has higher tendency than RHP6 to experience inter-chain interactions and self-assembly. This means that RHP7 may not interact with OmpT as strongly as RHP6 does due to the reduced availability of its hydrophobic segments. This observation agrees with our later discussion on diblock polymer as shown in Fig. 3E and highlights the importance to consider not only the abundance, but also the availability of RHP-protein interactions at the segment level.

11. *Fig. 3: The “No FBS”, “Native FBS”, and “Heated FBS” conditions are inconsistent with the main text discussion and remain unclear from the caption. Which test condition(s) contain RHP?*

Author response: Thank you for bringing this to our attention. In order to mediate any ambiguity, in all three conditions, we added various concentrations of RHP4 to test the cell viability of the NIH3T3 cell line. The discussion in the main text was based on observations examined in the “Heated FBS” condition, where native FBS and various concentrations of RHP4 were mixed, heated, and added into the cell culture media. The cell viability increased when more RHP4 was added prior to heating. To examine the effect of RHP4 alone on the cell viability, we performed the following two controlled experiments.

In the “No FBS” condition, RHP4 was the only additive in the cell culture media. When RHP alone with different concentrations was added in the cell culture media as shown in the black bar of Fig. 2E, the cell viability showed no significant change. In the “Native FBS” condition, native FBS and various concentrations of RHP4 were mixed prior to being added into the cell culture media. The cell viability slightly decreased as RHP concentration increased, as shown in blue bar. From the above two controlled experiments, we concluded that RHP4 has a trivial effect on cell viability and validated the current MTT assay as a feasible approach to test the activity of FBS on cell viability. Combined with the “Heated FBS” condition, we confirmed that RHP4 can protect the FBS’s activity, acting as a cell growth factor, upon heating.

To make the statement clearer, we’ve included a more detailed description of experimental conditions: “At given concentration of RHP4, RHP4 alone (No FBS), fresh FBS/RHP4 mixture (Native FBS), or heated FBS/RHP4 mixture (Heated FBS) was fed into the cell culture media.” in the caption of Fig. 2.

12. Lines 187–202: The authors’ discussion of LLPS is not technically accurate. LLPS can emerge as a bulk (non-microscopic) phenomenon; for example, liquid–liquid extraction due to the immiscibility of aqueous and organic phases is a large-scale application of LLPS. I encourage the authors to clarify that they are referring to microphase separation and/or coacervation, which can drive the formation of membraneless organelles in cellular systems. This discussion also requires copy-editing to meet the standards of technical publishing, including both a grammatical review and a closer look at the references. I noticed 2 separate sets of references in the main document, which appears to be the source of incorrect numbering throughout the LLPS discussion.

Author response: We thank the reviewer for pointing this out. We agree with the reviewer that the description was not completely accurate, so we have adjusted the language to make it clear that present studies focused on microscopic liquid droplets, a size scale of many membraneless organelles. This section has been copy-edited.

13. RHPs 8–12 should be compositionally defined in a supplementary table in similar fashion to RHPs 1–7.

Author response: As suggested by the reviewer, we have included information pertaining to RHPs 8–12 in the lower half of Table 1. There was a typo in the composition of RHP11 as the values in the OEGMA column and in the DMAEMA column were swapped. Correct values have been placed in the table. A note has been added in the main text to direct the reader to Table 1 as done for RHPs 1–7.

14. Line 209/Fig. S11: please include the equation used to fit the red line.

Author response: We think this is an excellent suggestion. We have added a note in the caption of Fig. S31 to direct the reader to the FRAP method section to find the equation, $R=A(1- e^{-(t/\tau)})$ (R is the recovery rate, A is the maximal recovery rate, and τ is the recovery half time).

15. Line 222/Fig. S14: It would be helpful to include RHP1 in this figure.

Author response: We fully agree and have added RHP1 into the figure accordingly.

16. Lines 225–231: *I do not understand the argument that “the right segments show up at the right place and right time.” Could the authors expand upon the current explanation?*

Author response: We thank the reviewer for pointing this out and made modification accordingly.

17. Line 233: *The authors note that RHPs form compact globules; however, a diameter of 8.8 nm seems on the large side for a 36 kDa biomolecule (if I recall correctly, proteins of this molecular weight would typically span 4–5 nm in diameter). I suggest the authors compare their SAXS data with scaling arguments for folded/ denatured proteins: <https://pubs.acs.org/doi/10.1021/bi991765q>*

Author response: Thank you for pointing this out. We agree with the reviewer’s comment that the average size of RHP’s particle measured by SAXS is intermediate between that of a folded protein and a denatured protein with the same molecular weight. On the other hand, the optical tweezers study suggests that RHPs adopt a continuum of conformations from compact globules to random coils. To be more precise, we changed the statement in the SAXS section to “solution small-angle x-ray scattering (SAXS) studies showed that RHP4 in water formed a single-chain nanoparticle, 8.8 nm in averaged diameter, to bury hydrophobic monomers.”

18. Line 235: *Visual snapshots from the MD simulation would support the arguments about RHP conformation and surface exposed vs. buried segments.*

Author response: We have previously reported that RHPs assemble with heterogeneous interfaces reminiscent of protein surfaces by molecular dynamics simulation (<https://pubs.acs.org/doi/full/10.1021/acs.macromol.0c01886>). To avoid duplicate discussion, we cited the paper and changed the sentence in the main text to: “The ¹H-NMR studies of RHP4 in D₂O showed that the surface-exposed segments are more hydrophilic and mobile; the buried segments are more hydrophobic and have a low tendency to snorkel to the surface of globular RHP chains (Fig. S16), consistent with previous atomistic molecular dynamic simulation”.

19. Fig. S20: *I don’t understand how these data suggest unraveling – could the authors elaborate upon this interpretation? I encourage including this data side-by-side or overlaid with the SAXS data in Fig. 5A.*

Author response: Thank you for pointing this out. We agree with the reviewer current analysis is not obvious and convincing and thus included additional analysis of Kratky plot ($Q^2I(Q)$ versus Q) to assess the globular nature of a polymer chain (<https://doi.org/10.1021/cr990071k>). Regarding the RHPs in DMSO, we decided to remove the data as present studies focus on the aqueous solutions of RHP ensembles as protein mixtures in biological fluids.

20. *The manuscript contains several typos and would benefit from minor corrections including but not limited to:*

- *Fig. 3B: “MTT Assay” (caption) should be defined.*

Author response: As the reviewer suggested, we have included in the caption a description of MTT assay: “The cell viability was indirectly measured by metabolic viability based assays (MTT assay) using tetrazolium salts, MTT ((3-(4,5-Dimethylthiazol 2-yl)-2,5- diphenyltetrazolium bromide).”

- *Fig. 3C: “Proteinase K” is misspelled.*

Author response: We thank the reviewer for pointing this out. We have decided to remove this panel to mainly focus on the discussion about FBS stability.

- *Table S3: Monomers 5 and 6 are both OEGMA (Mn=300 Da); I suspect one of these monomers should be Mn=500 Da.*

Author response: Thank you for pointing this out. We have corrected monomer 6 to be OEGMA (Mn = 500 Da).

- *Fig. 4A: I recommend labeling the sub-sub-panels (e.g., “Fig. 4A bottom left”) as separate panels.*

Author response: We think this is an excellent suggestion. We have modified Fig. 4 significantly in the revised manuscript.

- *Line 224: “Membraneless compartment” is misspelled.*

Author response: We thank the reviewer for pointing this out and have made the change accordingly.

- *Fig. S15: The yellow lines in the middle graph are difficult to read; is there a technical reason for the different colors used in the left and center plots?*

Author response: We thank the reviewer for pointing this out. We have decided to exclude this data from the revised manuscript.

- *Fig. 5D: “HLB” (y-axis) should be defined.*

Author response: Thanks for bring this to our attention. We have included a definition of HLB in the caption of Fig. 3 (please note that we have rearranged the order of figures in the revised manuscript) as, “The hydrophobicity of each monomer along a polymer chain was evaluated by the average hydrophilic-lipophilic balance (HLB) value of a sliding window.”

Reviewer Reports on the First Revision:

Referees' comments:

Referee #1 (Remarks to the Author):

I applaud the authors for taking the feedback constructively and revising the paper to address my concerns. I have no hesitation in recommending publication.

Referee #4 (Remarks to the Author):

The authors have sufficiently addressed my comments from peer review, and I recommend this manuscript for publication.